

# Updated Arctic melt pond fraction dataset and trends 2002–2023 using ENVISAT and Sentinel-3 remote sensing data

Larysa Istomina[1,2], Hannah Niehaus[2], Gunnar Spreen[2]

[1]Alfred-Wegener-Insitute, Helmholz Zentrum für Polar und Meeresforschung, Bremerhaven, 27570, Germany
[2]Institute of Environmental Physics, University of Bremen, Bremen, 28359, Germany

*Correspondence to*: Larysa Istomina (larysa.istomina@awi.de)

**Abstract.**

Melt ponds on the Arctic sea ice affect the radiative balance of the region as they introduce darkening of the sea ice during the Arctic summer. Temporal and spatial extent of the ponding as well as its amplitude reflect the state of the Arctic sea ice

and are important for our understanding of the Arctic sea ice change. Remote sensing retrievals of melt pond fraction (MPF) provide information both on the present state of the melt pond development as well as its change throughout the years, which is a valuable information in the context of climate change and Arctic amplification.

In this work, we transfer the earlier published Melt Pond Detector remote sensing retrieval (MPD) to the Ocean and Land Colour Instrument (OLCI) data onboard the Sentinel-3 satellite and so complement the existing Medium Resolution Imaging

Spectrometer (MERIS) MPF dataset (2002–2011) from Environmental Satellite (ENVISAT) with the recent data (2017–present). To evaluate the bias of the MPF product, comparisons to Sentinel-2 MultiSpectral Instrument (MSI) high resolution satellite imagery are presented, in addition to earlier published validation studies. Both MERIS and OLCI MPD tend to overestimate the small MPFs, which can be attributed to the presence of water saturated snow and sea ice before melt onset. Good agreement for middle range MPF is observed, and the areas of exceptionally high MPF = 100 % are recognized as

well.

The earlier published MERIS MPFs (2002–2011) were reprocessed using an improved cloud clearing routine and together with the recent Sentinel-3 data provide an internally consistent dataset, which allows to analyse the MPF development in the past 20 years. Although the total summer hemispheric MPF trend is moderate with +0.75 % per decade, the regional weekly MPF trends display pronounced dynamic and range from -10 % to as high as +20 % per decade, depending on the region.

We conclude on the following effects:

- the global Arctic melt onset shifted towards spring by at least 2 weeks, with the melt onset happening in late May in the recent years as compared to early-mid June in the beginning of the dataset.

- there is a change of the melt onset regime in the recent years, with East Siberian and Laptev Sea dominating the melt onset and not the Beaufort Gyre region as before.

- the Central Arctic, North Greenland and CAA show signs of increasing first year ice (FYI) fraction in the recent years.





The daily gridded MPF averages are available at the webpage of the Institute of Environmental Physics, University of Bremen, as a historic dataset for the ENVISAT data, and as ongoing operational processing for the Sentinel-3 data.

## 1 Introduction

The last 8 years, 2016–2022, have been the warmest in history (WMO, 2023), and the summer 2023 has been the hottest on
record (C3S, 2023). The Arctic is warming three to four times faster than the rest of the world (Rantanen et al., 2022) and the summer sea ice of today has reduced to half of its average extent of the 1980s (Perovich et al., 2020), it has become younger (Tschudi et al., 2020, Stroeve & Notz, 2018), and the floe residency time and ice thickness have been reducing as well (Haas et al., 2008; Sumata et al., 2023). Due to the sea ice being brighter than open ocean, both lateral and surface sea ice melt decrease the albedo of the Arctic Ocean during summer, affecting the energy budget and contributing to direct and indirect
surface albedo feedbacks within the Arctic amplification mechanisms (Wendisch et al., 2023).

According to the Global Climate Observing System (GCOS), sea ice is an Essential Climate Variable (ECV). The primary parameters of the sea ice ECV are ice concentration, area and extent, ice type, motion, deformation, age, thickness, and volume (GCOS-200, 2016). While the sea ice albedo has been included recently (GCOS-244, 2022), the sea ice surface melt, which is main contributor to the decreasing albedo and increasing transmittance of the sea ice in summer (Perovich et al.,
2002; Nicolaus et al., 2012, Light et al., 2022), is not yet considered a part of sea ice ECV.

Satellite remote sensing has been used to produce many of the sea ice ECV datasets to obtain pan-Arctic coverage (Sandven et al., 2023). However, the passive microwave (PM) sea ice concentration, which is used to produce the sea ice area and extent is compromised by summer sea ice surface melt (Ivanova et al., 2015; Kern et al., 2019, 2020, 2022). PM L-band and altimeter sea ice thickness datasets do not provide their products in the presence of melt ponds (Huntemann et al., 2014;
Patilea et al. 2019; Ricker et al., 2017), and only recently an altimeter-based sea ice thickness retrieval coupled with a machine learning approach has been presented (Landy et al., 2022). PM based sea ice drift product is affected during the melt season and is complemented with a parametric model during summertime (Lavergne et al., 2023). Not only the most drastic sea ice change happens during the Arctic summer, but it also is the most challenging season for the Arctic remote sensing retrievals.

Global climate models (GCM) have difficulty simulating melt ponds as well as they do not include sea ice topography, the major factor determining the melt pond fraction (MPF). At the time of writing, sea ice melt ponding has been included into the GCMs via parameterizations (Flocco et al., 2010; Hunke et al., 2013; Schroeder et al., 2014; Zhang et al., 2018). However, an agreement in terms of GCM melt pond representation is yet to be reached and the lack of thereof might explain parts of the existing discrepancies between the long term GCM sea ice forecast (Stroeve et al., 2012).

Although melt ponds have been observed *in situ* since many decades (Yackel et al., 2000; Perovich et al., 2001; Perovich et al., 2002; Eicken et al., 2004; Polashenski et al., 2012), the spatial and temporal coverage of these observations is sporadic.





There is a need for a climate conform remote sensing melt pond dataset which can be assimilated into the GCM and be potentially included into the sea ice ECV.

The currently published satellite melt pond datasets can be put in the following main groups: optical, passive microwave
(PM) and synthetic aperture radar (SAR). Among the optical MPF datasets, the spatial resolution defines the algorithm approaches depending on whether the melt ponds can still be detected separately or are already subpixel. The following sensor and algorithm groups can be distinguished: very high resolution (0.3–10 m, WorldView, Pleiades, commercial sensors: Wright and Polashenski, 2018; Webster et al., 2015), high resolution (10–60 m, e.g. Sentinel-2 MSI, Landsat-7,8: Rösel and Kaleschke, 2011; Wang et al., 2020; Li et al., 2020; Qin et al., 2021; Niehaus et al., 2023) and of moderate
resolution (250–1000 m, e.g. Moderate Resolution Imaging Spectroradiometer (MODIS), (Tschudi et al., 2008; Rösel et al., 2012; Ding et al., 2020; Lee and Stroeve, 2021; Feng et al., 2021; Peng et al., 2022), Medium Resolution Imaging Spectrometer (MERIS), (Zege et al., 2015; Istomina et al., 2015a)). In terms of swath width and revisit time, out of all optical sensors those of moderate resolution have the most potential to obtain daily pan-Arctic coverage. Sensors of higher resolution, although providing potential for high quality retrievals as the melt ponds are no longer subpixel, have limited
spatial coverage and can be used as ground truth for the moderate resolution retrievals. However, it must be noted that optical observations are hindered by clouds.

The PM sensor-based MPF retrievals (AMSR-E, AMSR-2 (Tanaka et al., 2016; Tanaka and Scharien, 2022), SMOS (Mäkynen et al., 2020)) do not have this disadvantage as they are only partly sensitive to the atmospheric influence at higher frequencies (89GHz, spatial resolution 3km). The coarse spatial resolution of lower frequencies (37 GHz–1.4 GHz,10–40
km) renders vast majority of data a subpixel mixture of lots of Arctic summer surface types. At microwave frequencies, the imaginary part of the complex permittivity of water differs by orders of magnitude from that of sea ice or snow, so that the penetration depth in snow and sea ice reduce drastically (from decimeters to submillimeters) in the presence of surface wetness and melt. So that PM MPF retrievals can only be used for dry cold sea ice surface with open melt ponds at 100 % ice concentration. When applied globally during Arctic summer, the resulting MPF will be biased as PM MPF retrievals
cannot distinguish between open water, water saturated surface and melt ponds.

The available SAR MPF retrievals (Scharien et al., 2017; Han et al., 2016; Fors et al., 2017; Li et al, 2017; Ramjan et al., 2018; Howell et al., 2020), in addition to limitations on the spatial coverage, are also affected by the inability to distinguish melt ponds and open water due to their equally low backscatter signal. In general case, an unknown sea ice surface roughness has to be resolved from an unknown water/melt pond surface roughness, and the angular backscatter dependency
delivers additional challenges in terms of signal-to-noise ratio. In some SAR scenes of higher resolution, however, single melt ponds can be detected by their shape.

Among optical/IR spectroradiometers, MODIS is the one with the longest time series of data available (Arctic data from MODIS Terra since 2000, MODIS Aqua since 2002). MODIS data is being comprehensively utilized to provide a variety of higher-level products in addition to the Level1B top of atmosphere (TOA) spectral reflectance, important for MPF retrievals
are the composite daily and 8-day cloud free surface reflectance products. After initial threshold- (Tschudi et al., 2008) and

fixed surface classes neural network approaches (Rösel et al., 2012), neural network MPF retrievals that use high resolution training data have followed (Ding et al., 2020; Lee and Stroeve, 2021; Feng et al., 2021; Peng et al., 2022). It is important to note the issues of the MODIS sensors such as the saturation over bright surfaces (Madhavan et al. 2012) and striping issue (Lee et al., 2020), which disturb the TOA reflectance and might affect the MPF retrievals. In addition, as the great variability

of the sea ice and melt pond inherent scattering properties is also affected by the past and future shifts in the Arctic sea ice properties, an adequate and versatile summer sea ice representation is required, so that limited training datasets used in the neural network approaches might not always suffice.

In this work, we present an MPF dataset based on an inversion of a physical forward model of snow covered and bare sea ice with melt ponds. This dataset is based on the algorithms described (Zege et al., 2015, Malinka et al., 2016, Malinka et al.,

2018) and consist of an improved version of earlier published historic MERIS dataset (Istomina et al., 2015a) and a new operational Ocean and Land Colour Instrument (OLCI) MPF dataset. As the MPD algorithm takes 9 spectral channels, we do not use MODIS due to the saturation issue mentioned above, and use MERIS (ENVISAT) and OLCI (Sentinel-3) instead. We perform comparisons to high resolution MPF data, investigate the internal consistency of the combined MERIS and OLCI dataset, and present Artic-wide MPF trends for 2002–2023 as an update to MPF trends 2002–2011 (Istomina et al.,

2015b).

## 2 Methods

The objective of this work is to continue the historic ENVISAT MPF dataset published earlier (Istomina et al., 2015b) using the validated MPD method (Zege et al., 2015; Istomina et al., 2015a) and Sentinel-3 OLCI data. As the optical sensors OLCI and Sea and Land Surface Temperature Radiometer (SLSTR) onboard Sentinel-3 are built to be direct successors of MERIS

and Along Track Scanning Radiometer (AATSR) onboard ENVISAT, the earlier published MERIS MPD retrieval can be applied. For this, we improve the earlier published cloud screening (Istomina et al., 2020) for the updated MERIS dataset, and use MERIS-consistent pre- and postprocessing routines also for the Sentinel-3 OLCI data.

### 2.1 Data used

The following remote sensing data have been utilized for the MPD retrieval:

- ENVISAT: operation time 2002–2012, Arctic data available 2002–2011, sensors MERIS (MPD MPF retrieval) and AATSR (training of the bayesian cloud screening MEris Cloudscreening Over Snow and Ice (MECOSI)).
- Sentinel-3A and 3B: operation since 2016 till present, Arctic data available since 2017, sensors OLCI (MPD MPF retrieval) and SLSTR (in synergy with OLCI - a cloud screening routine used as reference for MECOSI).

An overview of the spectral and spatial resolution characteristics of these sensors is given in Table 1.

To evaluate the quality of the obtained MPF dataset, we use Sentinel-2 MSI data for the OLCI dataset (Table 2), see Section 3 for details.



## 2.2 MPD retrieval

The MPD retrieval has been developed by Zege et al. (2015). The MPD algorithm takes Level-1B top of atmosphere reflectances at nine spectral channels as input (Table 1) and inverts the forward physical model of melting sea ice (Malinka

et al., 2016) with melt ponds (Malinka et al., 2018) to obtain the fraction of melt ponds in a given pixel as well as its black sky spectral albedo at 400, 500, 600, 700, 800, 900 nm which is then converted into broadband albedo according to Pohl et al., (2020). The sea ice is modeled as a stochastic medium and can represent various inclusions such as air bubbles, brine, and sediment. The parameters controlling the sea ice scattering properties are: optical thickness of the ice $\tau_{ice}$ , effective grain size of the scattering layer or snow cover $A_{eff}$, absorption coefficient of inclusions $\alpha_{inc}$. The melt pond is represented as a

Lambertian melt pond bottom of varying optical thickness $\tau_{pond}$ with the ice transport scattering coefficient $\sigma_{ice}$ and a freshwater layer of varying depth $\tau_{meltwater}$ on top. The atmospheric correction is performed using fast radiative transfer model (Tynes et al., 2001) for typical Arctic aerosol conditions (Tomasi et al., 2007). The constraints on the sea ice and melt pond model parameters are obtained from ~200 field spectra of sea ice and melt ponds measured by Istomina et al. (2013) during August 2012 in the Central Arctic. To perform the model inversion, we use the Newton-Rapson method (Press et al., 1992).

The resulting MPF is defined as the fraction of melt ponds divided by the fraction of ponded and not ponded sea ice. In case of open water present in the pixel, the resulting MPF deviates from this definition, see Section 3.2 for details.

The MPD have been validated against in situ, ship-based and airborne data (Istomina et al. (2015a). Case studies and trends for the MERIS dataset have been presented by Istomina et al. (2015b).

MPD does not perform cloud/surface classification of the input TOA reflectances and therefore relies on external cloud

screening and sea ice and snow flagging. The details of this procedure can be found in Section 2.3.

## 2.3 Cloud screening

A cloud screening routine of high quality is essential for the MPD retrieval as unscreened clouds contaminate the MPF product. Cloud screening over snow and ice is a challenging task as both clouds and the surface are bright and white in the visible spectral range. Near infrared (NIR) and thermal infrared (TIR) spectral bands have been proven to be more effective

for the task (e.g., Ackermann et al., 1998). As both MERIS and OLCI sensing range is limited to 900nm and 1020nm, respectively, we use synergy with, respectively, AATSR and SLSTR for TIR channels (Table 1).

For the MERIS part of the dataset, we use the Bayesian cloud screening MECOSI (Istomina et al., 2020). In MECOSI, a set of spectral and spatial features is utilized, using a VIS, NIR and TIR-based AATSR cloud mask (Istomina et al., 2010, 2011) as a reference dataset. In this work, we improve the reference dataset by omitting the equation (3) in Istomina et al., (2010),

as this snow flag correctly screens out higher melt pond fractions which must be preserved for this work. We also apply a threshold of 0.05 onto the reflectance component of the 3.7µm brightness temperature (BT) channel as described in Istomina et al. (2011) to help separate snow and ice from clouds. Then, the Bayesian approach is used to expand the reference AATSR cloud mask to the entire MERIS swath, as AATSR swath only covers one third of the MERIS swath. The resulting



swathwise cloud mask is applied to the MERIS swathes and the pixels which are cloud free during at least one overflight are
included (as opposed to areas consistently cloud free throughout the entire day in Istomina et al., 2020).

For the OLCI part of the dataset, as SLSTR swath covers OLCI swath completely, we use the MECOSI reference mask routine directly on the SLSTR and OLCI data. As SLSTR is the AATSR successor, no adaptation is needed.

During the SLSTR sensor recalibration, the TIR channels are either unavailable or of degraded quality (e.g., 300 ° K over snow and ice in the Arctic), so that the TIR part of the cloud screening cannot be used. In these cases, we alternate between
the Sentinel 3A and 3B platforms for a given day.

## 2.4 Daily gridded product

The adaptation of the MPD retrieval to the OLCI data comprises of accounting for the data format differences, producing synergy with the SLSTR data and establishing the operational processing. We use C Foreign Function Interface (CFFI) python package to write the wrapper on the MPD retrieval which is written in C++ programming language. OLCI data
preprocessing is done with GPT tool of the Brockmann Consult Java-based SNAP software.

Per default, Sentinel-3A data is used due to its longer dataset starting 2017 as compared to 2018 for Sentinel-3B. In case SLSTR or OLCI data discontinuity occurs for the given platform, Sentinel-3B data for that entire day is used, to produce possibly consistent daily averages of the MPF. Typically, 13–15 OLCI swathes per day are processed, with about 5 SLSTR granules corresponding to each OLCI swath subset.

The cloud screened MERIS/OLCI swathes are gridded into a 12.5km polar stereographic grid and stored as NetCDF files available for download. The OLCI and SLSTR files used for a given swath or a given daily average are stored in the corresponding NetCDF files as metadata for future reference. The minimal amount of cloud free OLCI pixels to form a valid 12.5km grid cell is $N_{valid\_OLCI} > 50$. As we exclude the dark pixels with the $R_{TOA\_412.5nm} < 0.3$ already earlier during the swath data processing, by limiting the amount of valid OLCI pixels during gridding, we exclude residual pixels of darker ice just
above the threshold, e.g., in an otherwise ice-free area. In addition, we remove an edge of 2 pixels on the cloud free areas in the swath data before gridding into the daily average. In this way, we preserve larger areas of continuous coverage, but remove single pixels or data with pixelwise gaps, as we expect these situations to occur in the areas where MPD cannot be applied (broken ice, slush ice, ice edge with lower SIC, cloud shadows around cloudy areas). An example of the daily product for the melt onset (13.06.2022, left) and for the height of the melt season (right, 27.07.2022) is shown in Figure 1.

## 3 Quality assessment of the resulting dataset

The MPD algorithm has been designed for use in areas of high sea ice concentrations with bare or snow-covered dry sea ice covered with blue melt ponds (Zege et al., 2015, Istomina et al., 2015a). When applying the algorithm to global data, deviations from this scenario are possible and potential bias needs to be investigated. In Section 3.1, we evaluate the quality of the MPD product for a range of the MPF values using comparisons to Sentinel-2 ground truth data.


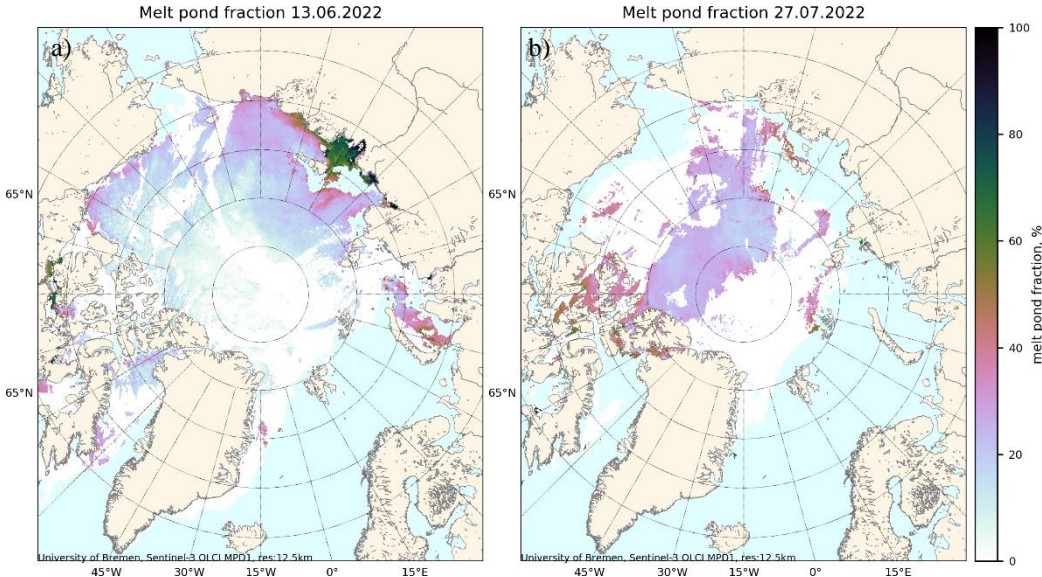

**Figure 1:** Example of the daily gridded MPD product for the 13.06.2022 and the 27.07.2022. Note the variable MPF on 13.06.2022 during the melt onset and the uniform MPF during the evolution of melt on 27.07.2022. The white areas depict the cloud covered sea ice where no MPF retrieval is possible. Sea ice coverage is given by ARTIST Sea Ice (ASI) PM sea ice concentration product (Spreen et al., 2008).

### 3.1 Evaluation of the full resolution swath MPF dataset

The MPD algorithm is based on the physical forward model of sea ice and melt ponds with boundary values for the parameters of air bubbles, brine inclusions and pollutants derived from field data. Its potential to account for the geophysical variability of the Arctic sea ice has been confirmed on *in situ* data (Malinka et al, 2016; Malinka et al., 2018). However, a correct interpretation of the sea ice scattering parameters from TOA reflectances of subpixel sea ice combined with melt ponds is not a trivial task. While there are satellite retrievals of e.g., effective grain size $A_{eff}$ of snow and sea ice (e.g. Wiebe

et al., 2013), the knowledge of sea ice inherent scattering properties (e.g. spectral extinction coefficient or better the transport scattering coefficient $\sigma_{ice}$ as used in MPD) is limited (Perovich, 1996).

The comparisons of the MERIS MPD algorithm to the point measurements on in situ, airborne and shipborne data has been performed by Istomina et al. (2015a), and comparisons to very high resolution (VHR, 1m pixel size) satellite data were presented by Marks (2015). When applying the MPD to the moderate resolution optical data like that of MERIS or OLCI,

subpixel mixture of many surface types occurs, and high resolution (HR) satellite imagery can aid in correct upscaling of the ground truth onto the global scale. As the spatial resolution of HR data is still lower than that of *in situ*, aerial photographs or VHR data, a classification and retrieval routine of its own is necessary to obtain the ground truth dataset.

In this work, we perform OLCI MPD comparison to 10m resolution Level-1C orthorectified TOA reflectances of Sentinel-2 MSI. For the evaluation of the MPF from the Sentinel-2 MSI imagery, a classification algorithm by Niehaus et al. (2023) is

used. In this algorithm, the difference between the spectral bands of wavelengths 490nm and 842nm within ice surface types



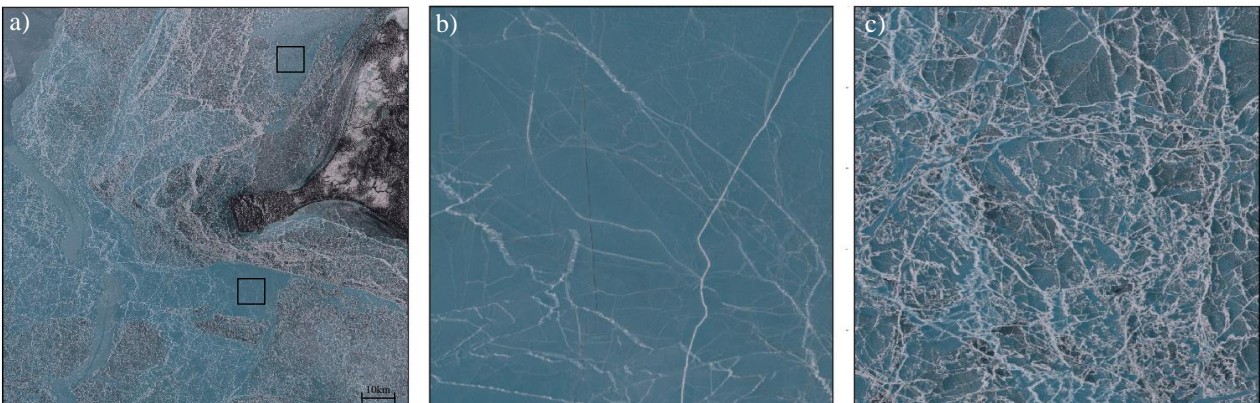

**Figure 2**: (a) RGB of the first full-resolution comparison case on 01.06.2021, MSI tile T54XVG, UTC Time 03h05′49″, relative orbit = R075, at landfast ice in the Laptev Sea, (b) RGB subset for the bottom square on (a), (c) RGB subset for the top square on (a).

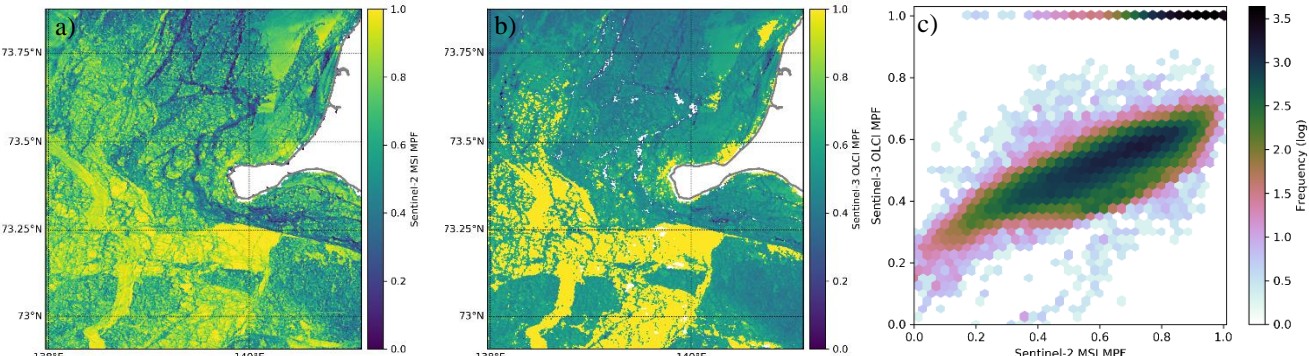

**Figure 3:** (a) MSI MPF for the case in Fig. 2a, (b) corresponding OLCI MPF, OLCI granule from 01.06.2021, time UTC 01h31′48″, cycle number 072, relative orbit 231, 300m resolution, (c) density plot of OLCI MPF correspondence to MSI MPF. Note the bimodal OLCI MPF for MSI MPF = 1.

is exploited (Grenfell et al., 1977) as adaptation of the LinearPolar Algorithm (Wang et al., 2020) for larger areas. Here, the resulting melt pond fraction is assigned under the assumption of linear mix between fixed sea ice and melt pond principal

axes. To ensure geospatial variability of the validation data, we use a comprehensive MPF dataset presented by Niehaus et al. (2023), with an addition of exceptionally high MPF on the landfast ice around Tiksi Bay in Laptev Sea, which is also presented as a case study below. In total, 50 scenes from June–August 2017–2023 are used for the evaluation (Table 2).

The resulting Sentinel-2 MSI MPF is downsampled to 300m for comparison to the full resolution Sentinel-3 OLCI MPF, and to 12.5km for comparison to the daily averaged OLCI MPF.

For comparisons at full resolution, we have selected two MSI tiles both observed on the 1st of June 2021: T54XVG and T53XMD, relative orbits R075 and R004, respectively.

In the first comparison case, an exceptionally high melt pond occurrence on the landfast ice took place (MSI tile T54XVG, orbit R075, Fig. 2a). The typical MPF values on first year ice (FYI) are assumed to be ~20–40 % during the height of melt season and up to 80 % during the melt onset peak, where lateral meltwater transport is responsible for the high MPF (Eicken





et al., 2004; Polashenski et al., 2012). An example of this process is the presented case where Sentinel-2 MSI detects a continuous field of MPF = 100 % stretching at least over 80km (Fig. 3a). The Sentinel-3 OLCI MPF also detects the area of 100 % MPF, showing good agreement (Fig. 3b) to MSI MPF. MSI MPF in the range 0.6–0.7 are slightly underestimated by the OLCI MPD retrieval (Fig. 3c), e.g., for the case shown in Fig. 2c. The two retrievals agree with the correlation coefficient R = 0.778 and root mean square deviation RMSD = 0.147. Bimodal OLCI MPF behavior in the high MSI MPF is visible (Fig. 3c, see upper MSI MPF range ~0.9). As the subpixel mixture of melting sea ice and melt ponds is spectrally ambiguous, the MPD retrieval utilizes two different solution families for these similar conditions, with the retrieval changing the ice and melt pond scattering parameters without changing the corresponding fractions as long as the boundary conditions allow. The jump towards MPF = 1 happens when the boundary condition onto the transport scattering coefficient of ice under the melt pond $\sigma_{ice} = 5$ is reached. Fig. 2a shows MSI RGB for the area of MPF = 1, where MPD MPF and MSI MPF

agree well (Fig. 2a, lower square, also see Fig. 3a and b). Here we see a continuous field of uniform blue ice with melt water on top.  The area where higher MSI MPF of 0.8–0.9 have been underestimated by OLCI MPD (MPF = 0.7) is shown on Fig. 2c and on Fig. 2a with the top square. Here, the sea ice surface is not as level: high fraction of ridges with accumulated snow and bright features on top of the melt ponds can be observed. In case of fresh snow ($A_{eff}$ ~50 µm), the NIR (>700nm) feature of the snow grain size causes an increase in the TOA reflectance as compared to larger grain sizes (e.g., Burkhart et al.,

2017), mimicking an increased subpixel fraction of melting sea ice ($A_{eff}$ ~500 µm and greater) and causing the MPF underestimation. The effective grain size (mean photon path length in a stochastic medium as modeled by MPD) at the retrieval convergence for the misclassified MPF (top square) is $A_{eff} = 1600$µm and $A_{eff} = 2300$µm for the correct MPF (bottom square) with corresponding optical thicknesses of white ice $\tau_{ice} = 53$ and 10. This confirms the assumption of the fresh snow presence under an assumption of otherwise equal sea ice properties. Subpixel fresh snow on partially frozen over

melt pond would likely not be visible in MSI MPF but can potentially be detected within MPD. Additionally, the inclined topography of the ridges might also bias the MPD retrieval as it distorts the directional reflectance properties of the (flat) sea ice surface assumed within MPD.

The second comparison case (MSI tile T53XMD, orbit R004, Fig. 4a) shows moderate MPF ~0.4 which agrees well to OLCI MPF, with areas of low MPF between 0 and 0.1 being overestimated by OLCI MPD (Fig. 5a, 5b). The good agreement at

midrange MPF changes into OLCI MPF displaying bimodal behaviour for the lower MSI MPF (Fig. 5c). Analysis of the MSI RGB for the area of good correspondence (Fig. 4b or left square on Fig. 4a) shows white ice with light blue melt ponds, the conditions for which the MPD retrieval has been designed. The area of MPD misclassification, where lower MSI MPFs were overestimated by OLCI, is shown in Fig. 4c and in Fig. 4a with the right square. Here, MSI MPF is less than 0.1 whereas OLCI MPF is 0.2. There are no visible melt ponds on top of the sea ice in Fig. 4c, but a darker water saturated sea

ice without thick snow cover or scattering layer, possibly subnivean ponds with meltwater already gathering on top of the sea ice but still beneath the snow cover. The spectral reflectance of sea ice surface just before melt differs only in amplitude but not so much in spectral shape from that of melt ponds (Istomina et al., 2013) so that misclassifications can occur due to the spectral ambiguity of the ice/pond mixture. The MPD grain size for this misclassification case (right square) is $A_{eff} = 500$ µm





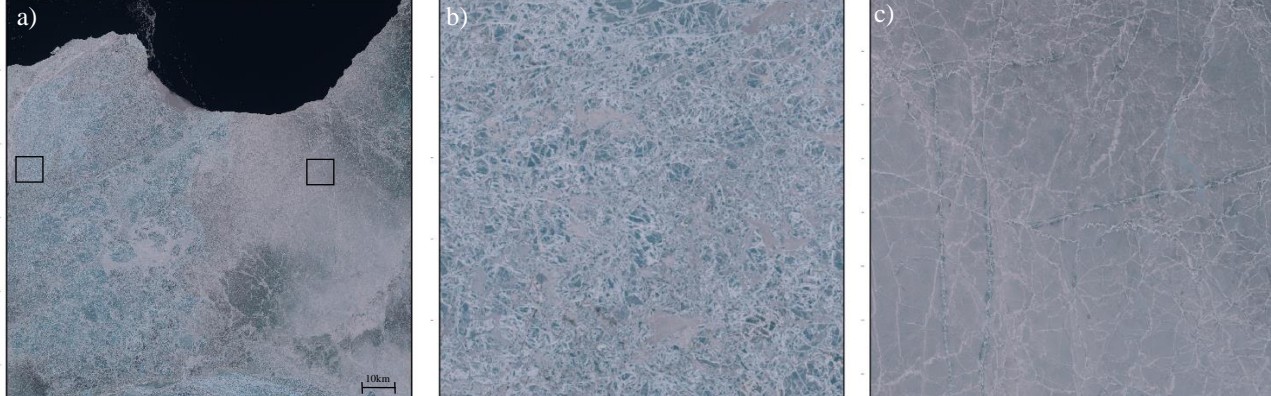

**Figure 4:** (a) RGB of the second full-resolution comparison case on 01.06.2021, MSI tile T53XMD, UTC Time 03h35′39″, relative orbit = R004. (b) RGB subset for the left square on (a), (c) RGB subset for the right square on (a).

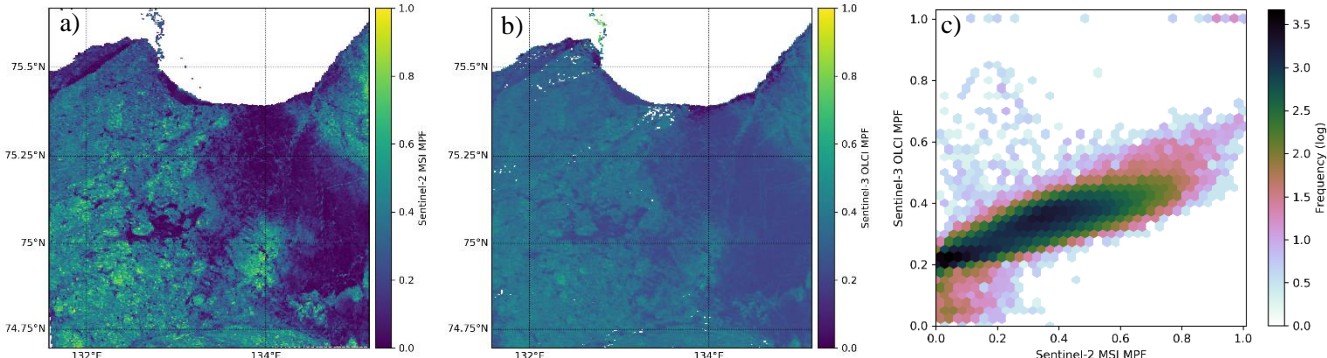

**Figure 5.** (a) MSI MPF for the case in Fig. 4a, (b) corresponding OLCI MPF, OLCI granule from 01.06.2021, time UTC 01h31′48″, cycle number 072, relative orbit 231, 300m resolution, (c) density plot of the OLCI MPF correspondence to MSI MPF. Note the bimodal OLCI MPF for MSI MPF < 0.1.

whereas the correctly retrieved case (left square) shows $A_{eff}$ = 1500 µm (white ice with light blue melt ponds). The boundary $\sigma_{ice}$ = 5 is reached for the left square but not the right ($\sigma_{ice}$ = 3.29), and $\tau_{ice}$ = 25 for the correct classification and is lower $\tau_{ice}$ = 10 for the misclassified case. The MPD retrieval appears to alternate between fine snow grains of fresh snow and high absorption of the water saturated sea ice underneath, which supports the assumption of translucent scattering layer or snow on top of this blue ice, with both surfaces influencing the OLCI TOA reflectance. The two retrievals agree with R = 0.85 and RMSD = 0.132. Further investigations are needed to clarify this behaviour of the MPD retrieval, especially with respect of investigating the boundary conditions (Malinka et al., 2016) and improving the Newthon-Rapson inversion routine (Zege et al., 2015). The blue ice presence affects the energy balance and can be utilized in future studies given adequate flagging of this surface type in the MPD MPF product.

The presented comparisons of the OLCI MPF against MSI MPF (Fig 3c and 5c) resemble the comparisons of the MERIS MPF to Global Fiducials Library (GFL) imagery (Marks, 2015), with good agreement of the values in the middle range, but overestimation of the lower MPF range and underestimation of the higher MPF by MPD. As in the case of GFL comparison,

this can be explained by the ambiguity of the sea ice-melt pond mixture, where inherent scattering properties of sea ice and melt pond are rather being varied to reproduce the TOA reflectance without changing the corresponding sea and melt pond

fractions far enough. A new feature of the presented comparison, namely the good agreement for MPF = 1, was not analysed for MERIS MPD for the absence of corresponding HR satellite data for the MERIS dataset. The exceptionally high MPF occurrence is, however, present in the OLCI and MERIS MPF also for other years (e.g., for 2022 in Fig. 1a), with the increasing tendence in the recent years (see Section 4 for corresponding MPF trends).

## 3.2 Evaluation of the daily gridded MPF dataset

To evaluate the quality of the daily gridded product, the 50 MSI scenes are downsampled to the 12.5km grid and compared to the daily OLCI MPFs (Fig. 1). Unfortunately, due to the Sentinel-2 MSI only observing coastal areas, Central Arctic and typical multiyear ice (MYI) areas are not represented. Nevertheless, the entire MPF range is present. The comparison scatter plot and the corresponding Sentinel-2 MSI data distribution are shown in (Fig. 6). Also, see Table 2 for details on the used MSI data. In this dataset the sea ice type is not exclusively landfast, so that open water fraction (OWF, OWF = 1 - SIC)

might affect the OLCI MPD retrieval. This is observed in Fig. 6a, where the majority of the MPD strong overestimation in the lower MPF range can be explained by a subpixel OWF (colored points). The grid cells with OWF = 0 display similar behavior as presented in the case study above, with a characteristic overestimation of small MSI MPF < 0.1 and underestimation around MSI MPF >0.7. Overall, the two datasets show good agreement with R = 0.86, sample size N = 3152, RMSD = 0.13, intercept = 0.17 and slope = 0.63.

It is important to note the offset of the OLCI MPF for the lower range of MSI MPF< 0.1. Although this dynamic is persistent

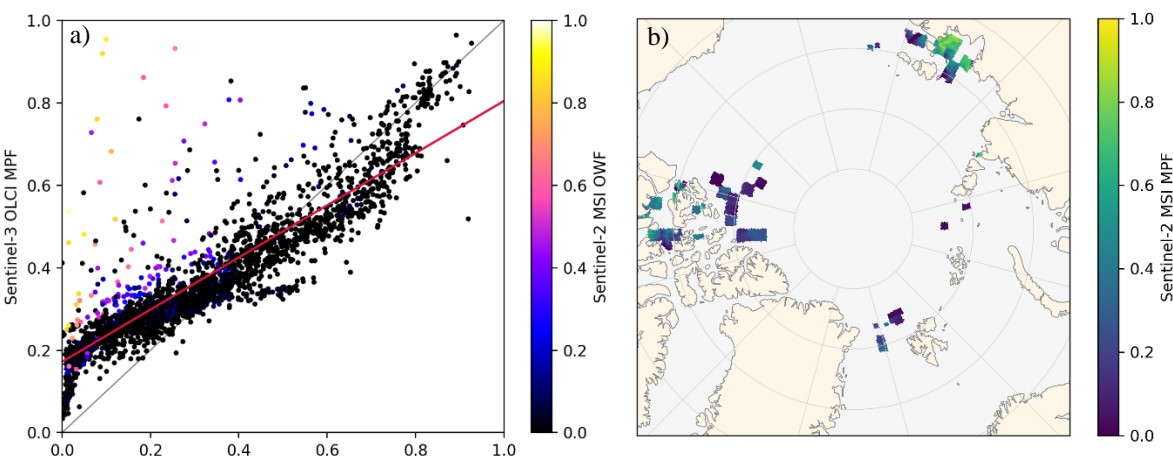

**Figure 6:** Evaluation of the daily gridded OLCI MPF dataset. (a): the comparison of the MSI MPF to OLCI MPF in relation to the fraction of open water as seen by MSI, R = 0.86, N = 3152, RMSD = 0.13, intercept = 0.17, slope = 0.63. (b): the spatial distribution of the MSI validation data used in (a), sensing time 2017−2023, see Table 2 for details on the used MSI dataset.




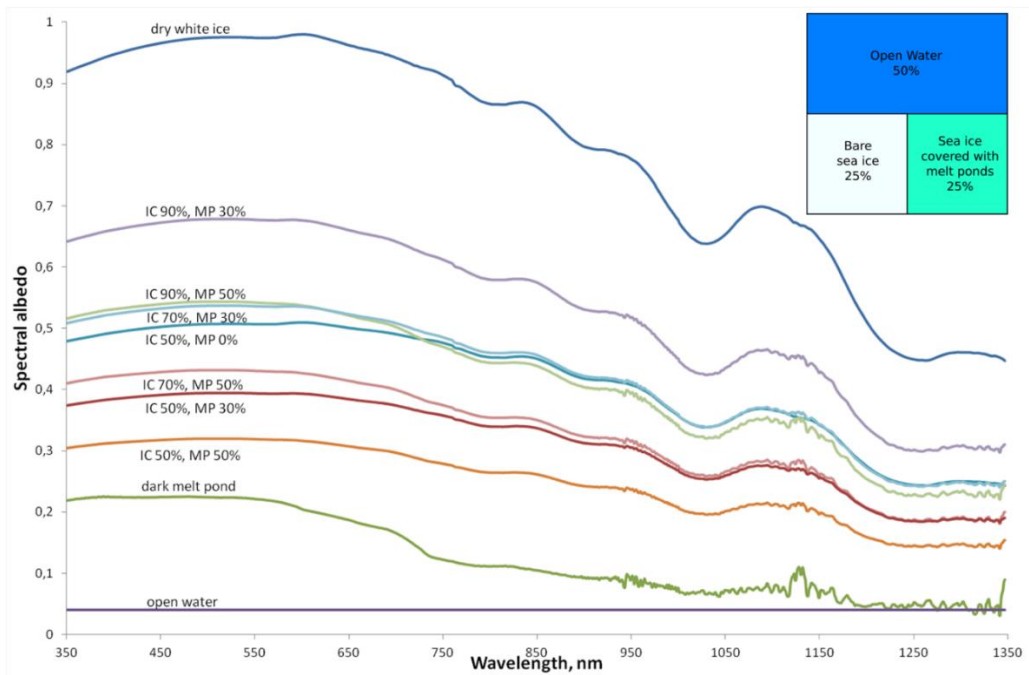

**Figure 7**: Linear mixture of in situ measured spectra of white ice and dark melt pond as the most frequent Arctic scenario. Moderate resolving spectroradiometers with optical and NIR bands will not be able to distinguish the influence of MPF from the influence of OWF for sea ice concentrations < 90 %. Spectral data for dark melt pond and dry white ice is taken from Istomina et al. (2016), open water is assumed to have a constant spectral albedo of 0.04.

throughout the entire validation effort also for MERIS (Zege et al., 2015; Istomina et al., 2015a; Marks, 2015), the pan-Arctic maps show MPF values in the range 0.01–0.05 regularly (e.g., Fig. 1, Fig. 9a). A possible reason for the observed discrepancy in the absence of open water is the discussed above misclassification of the water saturated sea ice for melt ponds surrounded by a surface with fine $A_{eff}$ (Fig. 5) as these surface classes are spectrally ambiguous in a subpixel situation. This spectral ambiguity occurs also for the open water and is illustrated in (Fig. 7).

Here, we mimic the Arctic conditions and mix various fractions of the three surface types: open water, bare white ice and dark melt pond, representing the frequent ice and melt pond types (spectra taken from Istomina et al., 2016), and mix them linearly with various fractions. The resulting spectra are shown in Fig. 7. The scenarios with SIC 50 %–90 % and MPF 0 %–50 % (green and blue lines in Fig. 7), as well as MPF of 30 %–50 % with SIC 50 %–70 % (red lines in Fig. 7), are challenging to distinguish correctly given the coarse spectral resolution of the moderate resolution spectroradiometers like MERIS, OLCI, but also MODIS, VIIRS etc. Figure 7 presents only one spectrum for each sea ice and melt pond types for the sake of clarity; given the great *in situ* surface type variability (Istomina et al., 2013, 2016), sea ice and melt pond are each represented by partly overlapping families of spectra, with an addition of surface types such as blue ice, drained melt pond, young ice, etc. Therefore, in the absence of additional information, the influence of the subpixel open water onto the retrieved MPF is virtually impossible to resolve and would cause mutual misclassification of open water and melt ponds and





vice versa. The result are falsely interdependent melt pond and open water classes, as shown for the case of neural network MODIS MPF retrieval (Rösel et al., 2012) by Marks, (2015, Section 4.4 and Fig. 4.27 therein). It is important to note that

this misclassification can potentially occur also in the areas of SIC = 100%, thus biasing even an otherwise favourable for the MPF retrieval situation.

To avoid this and to preserve the quality of the MPF MPF in the areas of 100% SIC, we refrain from separating the observed $R_{TOA}$ into three surface classes, and therefore expect some MPF overestimation in the areas with lower SIC, as is shown in Fig. 6a. Here, the SIC is color-coded as the colour of the data points. It can be seen, that low SIC < 50 % causes a strong

overestimation of the MPD MPF, especially for the cases of low MSI MPF and bright sea ice surface. This confirms the issue of the spectral ambiguity presented in Fig. 7 (light blue and green lines). On the other hand, cases with higher SIC > 70 % are within +0.05 MPF corridor from the regression line, reaching +0.2 MPF for SIC up to 50%.

From this we can conclude, that the effect of open water onto the MPD MPF is not linear and depends on whether or not the spectral ambiguity of the ice/water mixture can still be accommodated by changing the inherent ice or melt pond scattering

properties during the MPD algorithm iterations: in cases of higher SIC i.e. lower OWF, open water can be accounted for by using darker sea ice with larger grains, so that the resulting MPF is not affected, whereas in cases of higher OWF, the MPF has to be increased as well, as the boundary conditions do not allow for even darker sea ice. It has to be noted that, although the earlier (Zege et al., 2015; Istomina et al., 2015a and 2015b) as well as the current MPD version give the melt pond fraction of the pixel for the OLCI/MERIS swath data (Fig. 3, Fig. 5) as the fraction of open water cannot be accounted for,

the daily gridded product (Fig. 1) can be considered MPF of the *ice* fraction of the grid cell as the open water and low SICs have been removed during gridding, and only the relative MPF, i.e. MPF as a fraction of sea ice, is delivered. Marks (2015) has confirmed this by comparing to the product by Rösel et al. (2012) and showing good correspondence in case of the relative MODIS MPF (Fig. 4.21 in Marks, 2015).

As moderate resolution optical data alone is not sufficient to retrieve both SIC and MPF simultaneously (Fig. 7), we

recommend using an independent SIC product for the SIC evaluation of a given grid cell. It might stem from higher resolution optical data so that the open water and melting sea ice are no longer subpixel, or, depending on the required date range, PM SIC. Although PM SIC products are compromised in summer in the presence of surface melt, a recent study by Rostosky et al., (2023) suggests that SIC by the National Snow and Ice Data Center (NSIDC) (Meier et al., 2021) performs best even in the presence of surface metamorphosis associated with warm air intrusions. We therefore expect NSIDC SIC to

be less affected by the water saturated sea ice right before melt and therefore be potentially applicable up to the melt onset. In this work, however, we preserve the earlier published procedure (Istomina et al., 2015a) and do not account for the SIC of the grid cells. We thus expect MPF overestimation connected to the open water influence to be present and to play an increased role at the end of the melting season (depending on latitude, August–September), when cases of SIC < 70 % become spatially more frequent. Conversely, as the MPD MPF is not so much affected by SIC between 70 %–100 %, we

expect good performance of the MPD MPF product in the first half of the melting season (June–July).

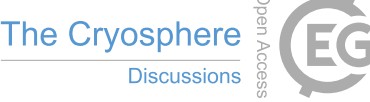



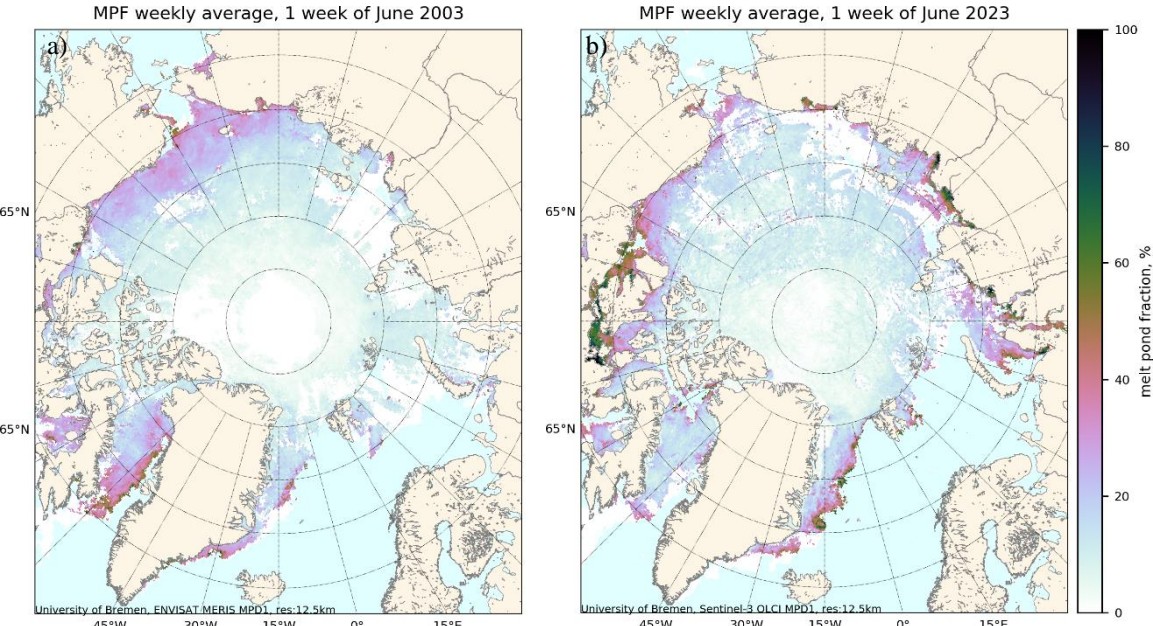

**Figure 8**: Weekly averages for (a) the first week of June 2003, MERIS MPD and (b) the first week of June 2023, OLCI MPD. Note the MPF differences in the Canadian Arctic Archipelago and Kara and Laptev Sea.

## 4 Weekly MPF trends

To produce the MPF trends, we average the daily gridded MPF into a 7-day average for each pixel of the NSIDC polar stereographic grid and analyse the resulting pan-Arctic maps for 2002–2023. An example of a weekly average is shown in Fig. 8 for the first week of June 2003 and 2023. In 2003, higher MPF was observed in the western Beaufort and Chukchi Sea in the beginning of the melting season, whereas in 2023 higher MPF can be seen the Kara, Laptev Sea and eastern Beaufort Sea and Canadian Arctic Archipelago (CAA). Note the very low MPF values in the high Arctic. This good performance of the MPD retrieval is observed throughout the entire dataset before the melt-associated surface darkening occurs (Fig. 9a).

The internal consistency of the combined dataset can be seen from the absence of offset, similar MPF minimum and maximum values, and similar MPF distributions between the MERIS and OLCI (Fig. 9). Here, MPF weekly averages for the 3 example weeks are shown: 1 weeks of May, June and July. August and September data tend to have limited spatial coverage due to higher cloud fraction during this time and are not shown. Note the very low MPF values for the first week of May with the mean MPF<0.1. The MPF values of 0.05 occur in the gridded product due to the misclassification of leads, as can be observed on the daily maps (Fig. 1a). The MPF data distribution for the first week of June (Fig. 9b and c) show increasing MPF with the stable MPF range from 0 to 1 and uniform MPF histogram. The first week of July presents further MPF increase with the majority of the data being greater than 0.2 and reaching MPF of 0.5 in the OLCI part of the dataset.



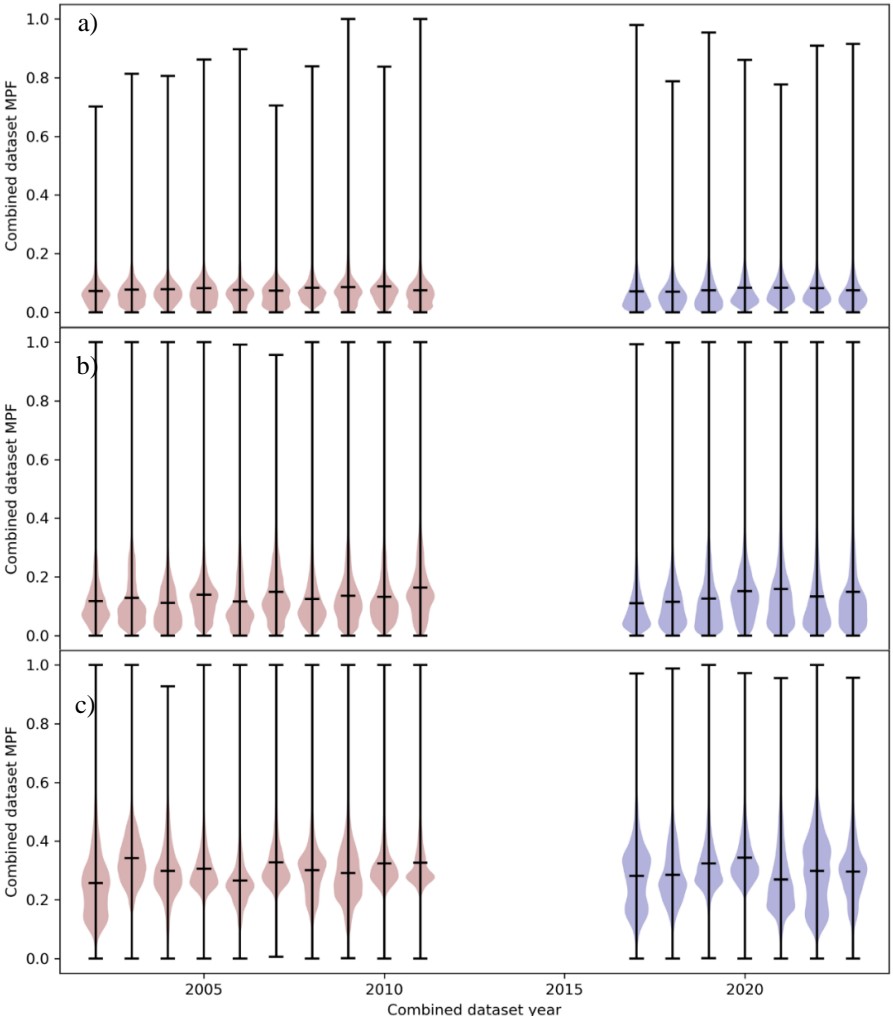

**Figure 9**: MPF distributions of the weekly averages for (a) the first week of May, (b) the first week of June, (c) first week of July of the combined MERIS (red) and OLCI (blue) dataset.

Out of 17 years of data, we take at least 11 valid points to produce a valid trend point. As weekly MPF averages do not have complete pan-Arctic coverage with arbitrary cloud gaps equally present in both parts of the dataset, we can employ this trade-off to obtain pan-Arctic coverage of the trend maps. Weekly trends from the fourth week of May till the fourth week of August (Fig. 10) are then produced via linear regression. Each week is then analysed separately, so that seasonality is eliminated. For a given week and grid cell, the MPF distribution is assumed to be near-normal throughout the dataset. As the grid cells were processed independently of each other, spatial continuity can be used to evaluate the quality of the trend, whereas the p-value (Fig. 11) is to be taken with caution given the small sample size and geophysical variability of the MPF.





## 4.1 Arctic MPF trend maps

The mean and maximum MPF on sea ice depends not only on the air temperature, but also on its roughness and other parameters (Polashenski et al., 2012). The Arctic sea ice consists of two main types: the FYI, featuring *uniform* surface with
larger maximum MPFs up to 80 % being possible, and MYI, where the *rougher* surface prevents high MPFs, with maximum values being only up to 30 % (Untersteiner, 1986; Perovich et al., 2002; Eicken et al., 2004). These different ice types display different temporal behaviour during the melt season. The FYI is going through 1. melt onset, 2. melt maximum of up to 80% MPF, 3. drainage, 4. evolution of melt with the lower second MPF peak and disintegration, and MYI experiences a single wide MPF peak in the middle of the melting season with little to no drainage and MPF up to 30% (Eicken et al., 2002;
Istomina et al., 2015a and Fig.1 therein). This dependence of MPF onto the sea ice type and its surface roughness needs to be considered while analysing the MPF trends. Similarly, with the recent change influencing the sea ice thickness (Sumata et al., 2023), the ability of sea ice to hold a certain MPF before drainage (Polashenski et al., 2017) may have been affected as well, and changes of precipitation affecting snow depth and available meltwater (Webster et al., 2014) cannot be excluded either. In the following trend discussion, all MPF trends are given in percent per decade.
A strong positive MPF trend reaching +20% in the **Kara and Laptev Sea** can be observed starting in the 4th week of May to the 2nd week of June (Fig. 10a–c) is followed by a spatially inhomogeneous negative trend up to -10 % in the 3rd and 4th weeks of June (Fig. 10d–e) and then turns into positive trend of +10 % during the 1st and 2nd week of July (Fig. 10f–g). We interpret this as a shift of the four FYI melt phases towards spring. The negative MPF trend occurs when the pond drainage occurs already in the week of the melt onset peak. Similarly, a melt phase temporal shift by at least 2 weeks towards spring
is observed in the Central Arctic, marked with the rectangle in Fig. 10c–g.

The negative trend around -6 % in the **Beaufort and Chukchi Sea** in the 2nd week of June (Fig. 10c) is preceded and followed by the spatially inhomogeneous trends of +2% (Fig10b and d), so that also here a temporal shift of melt phases can play a role. However, as the amplitudes of these trends do not match, an actual decrease of MPF in this area is possible, e.g., due to an increase in sea ice roughness or due to lower meltwater availability via a decreased snow depth in the western
Arctic (Webster et al., 2014). These factors would cause the MPF to increase gradually instead of a strong melt onset peak, so that higher MPFs do not happen till 3rd week of June. This assumption is supported by a spatially matching positive MPF trend of +6 % in Fig. 10g which corresponds to the peak of the gradual MPF evolution.

**South CAA** shows the temporal shift of the melt onset from the 4th week of June and 1st week of July (Fig 10e–f) to 1st–3rd weeks of June (Fig. 10b–d) with a positive trend of +12%. The positive MPF trend of +2% to +6% during the height of melt
season during the 3rd week of July though to the 2nd week of August (Fig. 10h–k) corresponds to the ice type shift towards FYI in this landfast ice area. Interesting to note is that the observed MPFs are higher than the typical pack FYI MPF. The MPF evolution in 2023 shows an initial melt onset on the 14th of June with MPF ~ 50%, an MPF decrease to 30 % on the 8th of July, and a higher second peak of MPF ~ 65% on the 29th of July 2023 and till the ice disintegration on the 15th of August. For comparison, in 2018, the MPF is ~ 40% throughout the entire season without much variation (Istomina, 2023b).





**Figure 10**: Weekly MPF trends of the combined MERIS and OLCI dataset 2002–2023 (data 2012–2016 not available).

The MYI area **north of Greenland** (marked with an oval, Fig.10d–e and Fig.10 k–l) displays an inhomogeneous MPF trend
of up to +6 % throughout the melt season with an increase at the end of August. This increase might indicate the ice type
shift towards FYI, but also open water influence on the MPF meaning a complementary negative SIC trend. Both sea ice
type shift and potential negative SIC trend are indicators of younger, more prone to break-up sea ice (Maslanik et al., 2007,
Gregory et al., 2022) in this typical MYI region.

The negative MPF trend between -1% and -4% in the **Central Arctic** in the height of the melting season for the 2$^{nd}$– 4$^{th}$
weeks of July (Fig.10g–i) can be interpreted as the ice type shift towards FYI, where the MYI melt peak is being replaced
with the FYI melt evolution phase. The FYI onset peak in the 4$^{th}$ week of May – 3$^{rd}$ week of June with the trend +3% (Fig.
10a–d) and the FYI drainage phase in the 4$^{th}$ week of June (Fig10e) seen as is the negative MPF trend confirm this
assumption. Subsequent negative MPF trend -5% in August (Fig. 10j–k) suggests an increased role of pond drainage
connected to the decreased sea ice thickness, so that average MPF is not as high as in the earlier years of the dataset.







**Figure 11**: Trend significance for the weekly MPF trends of the combined MERIS and OLCI dataset in Fig. 10.

It has to be noted that the displayed MPF trends are only valid under an assumption of absent cloud cover trend, i.e., irregularities of the Arctic cloud coverage throughout the combined dataset years will influence the MPF trend due to irregular representation of e.g. different melt stages. Similarly, we attribute at least some of the positive MPF trends to the presumably decreasing summer sea ice concentration trend, as the thinner, younger Arctic sea ice would be prone to sea ice motion and lead formation (Maslanik et al., 2007; Gregory et al., 2022).

## 4.2 Hemispheric averaged MPF trends

The weekly hemispheric MPF trends for the combined MERIS and OLCI dataset are shown in Fig. 12 and the corresponding values of trend in percent per decade, trend baseline and p-value are given in Table 3.



Despite pronounced regional MPF variability, the weekly hemispheric MPF trends are moderate in the range of +0.15 %–+3 %, except for 2nd and 4th weeks of June, where the negative trend can be attributed to the increased melt in the eastern Arctic and melting season shifting towards spring (Sect. 4.1). Significant hemispheric MPF trend of +1% is observed at the end of May (4th week) and beginning of June (1st week). Last three weeks of July display consistent trend of +0.6–+0.7 %, which

can be explained by higher MPFs during melt evolution stage on a flatter, younger ice in the recent years of the dataset, as opposed to lower MPFs on rough MYI in the beginning of the dataset. The positive trends of +1–+3 % seen in the last three weeks of August are potentially connected to the change of the ice type toward FYI as well, but are likely enhanced by the negative SIC trends associated with thinner sea ice being more prone to break-up. As can be seen from the regional dynamic (Fig. 10), this stands also for the MYI region north of Greenland and is statistically significant. The total hemispheric MPF

trend for the entire melting season from the 4th week of May till the 4th week of August is moderate with +0.75 % per decade. The weekly averaged hemispheric MPF displays positive dynamic for the summer 2023 as compared to the previous years, with $MPF_{2023}$ being in the top 20 percentile for 8 out of 13 weeks shown in Table 3 and displaying highest MPF of the combined dataset for 6 weeks (3rd, 4th week of May and July, and 2nd, 4th week of August).

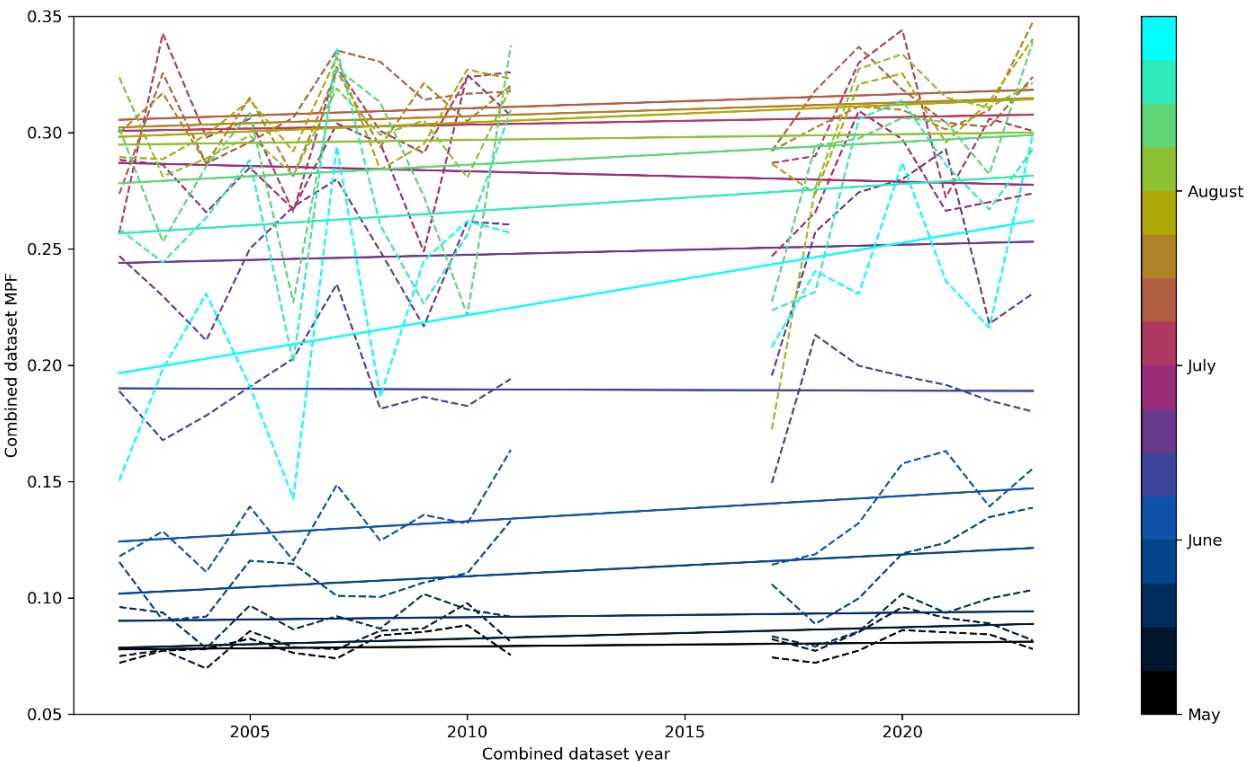


**Figure 12**: Weekly hemispheric MPF trends of the combined MERIS and OLCI dataset.

The earlier published MERIS dataset displayed positive MPF trends in the CAA and North Greenland MYI region in 2002-2011 (Istomina et al., 2015b), which are potentially caused by the loss of older, thicker sea ice (Maslanik et al., 2007;

Sumata et al., 2023) after 2007. The additional OLCI data 2017-2023 suggests the sea ice type change towards FYI for this and other regions, in addition to an earlier melt onset where the East Arctic predominates. Further investigations concerning the potential change of atmospheric and ocean circulations, which can play a role e.g., via the Arctic Oscillation (e.g., Lim et al., 2022), are needed to further clarify the observed MPF trend variability.

**5 Summary**

Melt ponds play a key role in the energy balance of the sea ice covered Arctic Ocean during summer. In order for the summer sea ice melt to be included in the climate models, long term remote sensing datasets are needed. In this work, we present a combined remote sensing melt pond fraction dataset produced from ENVISAT MERIS and Sentinel-3 OLCI sensors based on a physical forward model of sea ice and melt ponds. The resulting dataset ranges from 2002 and is ongoing within daily operational processing. We apply the earlier published for MERIS MPD algorithm on OLCI data and update the

cloud screening routine to ensure internal consistency of the combined dataset. We perform quality evaluation of the new OLCI dataset against high resolution Sentinel-2 MSI MPFs and analyse the MPF trends from 2002–2023, omitting 2012–2016 due to no available data.

Intercomparison studies between OLCI and Sentinel-2 MSI MPF show good correspondence for middle MPF range with an overestimation in the lower MPF range which is connected to the presence of water saturated snow and sea ice. Good

correspondence for the very high MPF = 100% is observed. The mean correlation coefficient from the full resolution and daily gridded comparisons to Sentinel-2 MSI is R = 0.84 and mean RMSD = 0.137.

As moderate resolution VIS/NIR data alone is not sufficient for simultaneous MPF and SIC retrieval due to the spectral ambiguity between subpixel melting sea ice and open water, the open water is not accounted for in a purely optical MPF retrieval. Within MPD, it introduces an overestimation of +0.05 MPF for SIC > 70 %, and up to +0.2 MPD MPF for SIC ~

50%. Threshold-based and morphological filters are applied to remove lower SIC in the daily gridded product, so that high quality MPF is expected for the first half of the melting season in June-July before the sea ice disintegration phase in August. The internal conformity analysis between MERIS and OLCI datasets showed good consistency with no systematic differences in the lower and higher MPF range as well as the MPF distribution shapes. Despite of the known effect of the water saturated sea ice and leads both of which will cause MPF overestimation, low MPF values MPF < 0.1 are consistently

seen for both datasets for the exemplary 1st weeks of May and June of the dataset.

Analysis of the weekly MPF trend maps showed pronounced regional variability with peak trend values between -10 % and +20 % per decade. Depending on the region, also moderate weekly MPF trends are observed in the range between -5% and +5% per decade.



The significant positive trend around 15% in the Laptev and Kara Sea combined with the spatially extended negative MPF trend of -6% in the Beaufort Gyre region in the beginning of the melting season lets us assume the melt onset regime shift in the recent years, where the Eastern Arctic dominates the melt onset and not the Western Arctic as in the earlier years of the dataset. The exceptionally high MPF in the Laptev Sea is confirmed with the Sentinel-2 MSI MPF for June 2021 and is visible also in the other years of the combined MPF dataset.

The observed melt onset shifted at least 2 weeks towards spring and signs of sea ice type change from MYI towards FYI are observed in the Central Arctic, CAA and North Greenland. The observed regional dynamics of the MPF trend suggests that, in addition to the ice relief determining the MPF, additional parameters like sea ice permeability and thickness, precipitation and meltwater availability need to be analysed to fully clarify the observed regional MPF trend dynamics.

Hemispheric averaged MPF trends display positive trends +0.15 %– +3 % per decade for all weeks except for the negative trends in the 2$^{nd}$ and 4$^{th}$ week of June, which can be partly attributed to melt stages shifting towards spring. This trend behaviour is likely connected to the increased role of thinner, younger sea ice on the pan-Arctic scale in the recent years.

We conclude that despite pronounced interannual variability, the Arctic MPF is in moderate long-term increase with the hemispheric MPF trend of +0.75 % per decade, with the summer 2023 advancing the positive MPF trend.

Additional studies are needed also to evaluate the effect of potential atmospheric and SIC trends onto the observed MPF trends.



Data availability.

The OLCI MPFs for 2017–ongoing are available at https://seaice.uni-bremen.de/data/olci/ (last access: 11 September 2023,
Istomina, 2023a). The historic MERIS MPF dataset is available at: https://data.seaice.uni-bremen.de/meris/mecosi/ (last
access: 11 September 2023, Istomina, 2023b).

Author contributions.

LI adapted the MPD retrieval and cloud screening routine to Sentinel-3 data, performed the comparison to the classified
Sentinel-2 MSI MPF data, established Sentinel-3 operational processing, performed the new training of the cloud screening
MECOSI and reprocessed the MERIS dataset, produced the updated MPF trends, and outlined the manuscript. HN
performed the MPF retrieval on the Sentinel-2 MSI data. GS provided the computational infrastructure and participated in
discussions of the results. All authors provided critical feedback on the manuscript and contributed to the text.

Competing interests.

The authors declare that they have no conflict of interest.

Acknowledgements.

The authors express gratitude to ESA and the EU for providing ENVISAT, Sentinel-2, Sentinel-3 data and to Brockmann
Consult for providing the software packages BEAM and SNAP.
This work has been funded as a part of EU project SPICES, German Research Foundation (Deutsche
Forschungsgemeinschaft, DFG) SPP 1158 project REASSESS and TRR 172 within the collaborative research project (AC)[3]
on Arctic Amplification.

Financial support.

This research has been partly supported within the EU project SPICES, grant agreement ID: 640161, by the DFG SPP 1158
project REASSESS, grant no. 424326801, and DFG TRR 172, Project-ID 268020496.

The article processing charges for this open-access publication were covered by the University of Bremen.



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





**Table 1: Moderate resolution spectroradiometers used in this work; spectral bands used in MPD retrieval are shown in bold *italic*.**

| Sensor acronym | OLCI | SLSTR | MERIS | AATSR |
|---|---|---|---|---|
| Swath width | 1270km | 1470km | 1150km | 512km |
| Resolution:full (reduced) | 300m (1.2km) | 500m (1km) | 300m (1.2km) | 1km |
| Spectral Channels | 400 | – | – | – |
| VIS (nm) | *412.5* | – | *412.5* | – |
| | *442.5* | – | *442.5* | – |
| | *490* | – | *490* | – |
| | 510 | – | 510 | – |
| | 560 | 555 | 560 | 555 |
| | 620 | *660* | 620 | *660* |
| | 665 | – | 665 | – |
| | 673.75 | – | – | – |
| | *681.25* | – | *681.25* | – |
| | 708.75 | – | 708.75 | – |
| | *753.75* | – | *753.75* | – |
| Spectral Channels | 761.25 | – | 760.625 | – |
| NIR (nm) | 764.375 | – | – | – |
| | 767.75 | – | – | – |
| | *778.75* | – | *778.75* | – |
| | *865* | *870* | *865* | *870* |
| | *885* | – | *885* | – |
| | 900 | – | 900 | – |
| | 940 | – | – | – |
| | 1020 | – | – | – |
| Spectral Channels | – | 1.37 | – | – |
| TIR (µm) | – | *1.6* | – | *1.6* |
| | – | 2.2 | – | – |
| | – | *3.74* | – | *3.74* |
| | – | *10.8* | – | *10.8* |
| | – | 12 | – | 12 |



**Table 2: Sentinel-2 MSI scenes used for comparison to Sentinel-3 OLCI MPF.**

| Date | Tile | Latitude | Longitude | Area, km$^2$ | MSI MPF(STD) | OLCI MPF(STD) |
|------|------|----------|-----------|---------|--------------|---------------|
| 03.07.2017 | T10XEN | 78.77 | -120.5 | 14687.5 | 0.275(0.053) | 0.292(0.028) |
| 05.07.2017 | T12XVL | 76.79 | -111.96 | 6718.75 | 0.38(0.057) | 0.371(0.025) |
| 10.06.2018 | T48XWM | 77.88 | 105.57 | 2812.5 | 0.628(0.117) | 0.566(0.142) |
| 25.06.2018 | T43XEM | 82.4 | 76.95 | 5781.25 | 0.023(0.013) | 0.19(0.018) |
| 28.06.2018 | T11XNJ | 79.65 | -114.34 | 15000.0 | 0.087(0.037) | 0.232(0.033) |
| 05.07.2018 | T12XWP | 79.71 | -109.07 | 10625.0 | 0.249(0.052) | 0.314(0.029) |
| 11.08.2018 | T57XVC | 74.63 | 157.64 | 2812.5 | 0.135(0.029) | 0.326(0.007) |
| 06.07.2019 | T14XML | 77.06 | -102.23 | 5781.25 | 0.408(0.068) | 0.363(0.029) |
| 07.07.2019 | T11XNF | 77.3 | -113.66 | 2656.25 | 0.581(0.099) | 0.489(0.041) |
| 10.07.2019 | T57XWD | 74.88 | 159.72 | 3437.5 | 0.344(0.029) | 0.319(0.007) |
| 30.07.2019 | T13XEL | 81.45 | -101.73 | 15156.25 | 0.294(0.029) | 0.315(0.022) |
| 05.08.2019 | T13XEM | 82.26 | -101.35 | 13125.0 | 0.269(0.02) | 0.297(0.014) |
| 21.06.2020 | T33XVM | 82.12 | 12.09 | 8750.0 | 0.024(0.006) | 0.18(0.012) |
| 22.06.2020 | T31XEM | 82.05 | 8.54 | 4687.5 | 0.015(0.005) | 0.184(0.009) |
| 30.06.2020 | T31XEL | 81.7 | 8.21 | 4218.75 | 0.168(0.032) | 0.266(0.015) |
| 01.07.2020 | T33XVL | 81.58 | 11.24 | 8593.75 | 0.096(0.022) | 0.236(0.007) |
| 05.07.2020 | T08XMQ | 80.37 | -138.43 | 6562.5 | 0.472(0.044) | 0.339(0.012) |
| 07.07.2020 | T31XEL | 81.59 | 4.02 | 2031.25 | 0.34(0.022) | 0.345(0.014) |
| 11.07.2020 | T13XEL | 81.45 | -101.73 | 15156.25 | 0.245(0.048) | 0.298(0.016) |
| 14.07.2020 | T12XWP | 79.61 | -107.99 | 5468.75 | 0.212(0.016) | 0.29(0.011) |
| 22.07.2020 | T30XWQ | 80.57 | -1.17 | 6875.0 | 0.215(0.027) | 0.299(0.019) |
| 27.07.2020 | T30XWP | 79.84 | -1.11 | 7656.25 | 0.329(0.054) | 0.388(0.032) |
| 06.08.2020 | T31XDL | 81.73 | -1.98 | 2812.5 | 0.238(0.017) | 0.311(0.019) |
| 10.08.2020 | T09XWK | 80.42 | -125.23 | 3281.25 | 0.165(0.015) | 0.262(0.009) |
| 10.06.2021 | T10XDM | 77.87 | -124.92 | 14843.75 | 0.011(0.006) | 0.08(0.017) |
| 17.06.2021 | T08XNR | 81.34 | -131.19 | 10000.0 | 0.026(0.017) | 0.142(0.032) |
| 04.07.2021 | T10XDQ | 80.56 | -125.42 | 14531.25 | 0.107(0.021) | 0.233(0.014) |
| 04.07.2021 | T11XMJ | 79.62 | -118.99 | 12968.75 | 0.129(0.025) | 0.258(0.015) |
| 19.07.2021 | T13XEK | 80.65 | -102.51 | 11093.75 | 0.15(0.03) | 0.269(0.01) |
| 19.07.2021 | T14XMQ | 80.64 | -101.49 | 13125.0 | 0.149(0.032) | 0.269(0.01) |
| 19.07.2021 | T45XVK | 80.4 | 86.14 | 5468.75 | 0.043(0.015) | 0.197(0.009) |
| 01.06.2021 | T55XEB | 73.34 | 148.45 | 11875.0 | 0.097(0.127) | 0.32(0.063) |
| 01.06.2021 | T52XEG | 73.39 | 130.76 | 14531.25 | 0.643(0.176) | 0.629(0.099) |
| 01.06.2021 | T52XEJ | 75.11 | 132.07 | 6093.75 | 0.131(0.173) | 0.381(0.044) |
| 01.06.2021 | T53XMD | 75.09 | 133.46 | 11406.25 | 0.237(0.174) | 0.361(0.077) |
| 17.06.2023 | T11XMD | 75.12 | -118.86 | 11562.5 | 0.284(0.269) | 0.581(0.176) |
| 31.05.2021 | T53XMC | 74.28 | 133.54 | 14062.5 | 0.38(0.126) | 0.372(0.066) |
| 31.05.2021 | T53XMD | 75.09 | 133.45 | 11562.5 | 0.108(0.112) | 0.267(0.061) |





| 31.05.2021 | T53XNB | 73.37 | 136.58 | 15625.0 | 0.451(0.155) | 0.443(0.08) |
| 01.06.2021 | T53XNA | 72.47 | 136.5 | 15625.0 | 0.748(0.069) | 0.67(0.111) |
| 01.06.2021 | T54XVF | 72.58 | 139.28 | 11250.0 | 0.543(0.319) | 0.725(0.117) |
| 01.06.2021 | T54XVG | 73.38 | 139.51 | 14375.0 | 0.603(0.188) | 0.583(0.123) |
| 01.06.2021 | T55XDB | 73.4 | 145.56 | 15156.25 | 0.252(0.183) | 0.388(0.065) |
| 18.06.2021 | T13XEC | 74.3 | -103.21 | 15468.75 | 0.238(0.116) | 0.29(0.068) |
| 18.06.2021 | T13XED | 75.15 | -103.08 | 14062.5 | 0.234(0.13) | 0.276(0.069) |
| 18.06.2021 | T14XMH | 74.4 | -100.73 | 12031.25 | 0.094(0.095) | 0.232(0.058) |
| 14.06.2023 | T12XVG | 73.41 | -112.36 | 14687.5 | 0.438(0.109) | 0.488(0.039) |
| 15.06.2023 | T12XWF | 72.63 | -109.28 | 7343.75 | 0.215(0.22) | 0.543(0.098) |
| 15.06.2023 | T13XEB | 73.41 | -103.14 | 13750.0 | 0.375(0.183) | 0.456(0.068) |
| 17.06.2023 | T11XND | 74.93 | -115.38 | 5312.5 | 0.256(0.277) | 0.66(0.078) |

805

**Table 3: Weekly hemispheric MPF trends of the combined MERIS and OLCI dataset 2002−2023 (data 2012−2016 not available).**

| Month | Week | Trend, % per decade | Trend baseline | p-value |
| --- | --- | --- | --- | --- |
| May | 4 | 0.93 | 0.10 | 0.07 |
| June | 1 | 1.09 | 0.12 | 0.06 |
| June | 2 | -0.05 | 0.19 | 0.94 |
| June | 3 | 0.43 | 0.24 | 0.66 |
| June | 4 | -0.45 | 0.29 | 0.56 |
| July | 1 | 0.33 | 0.30 | 0.72 |
| July | 2 | 0.61 | 0.31 | 0.32 |
| July | 3 | 0.60 | 0.30 | 0.29 |
| July | 4 | 0.77 | 0.30 | 0.27 |
| August | 1 | 0.25 | 0.29 | 0.85 |
| August | 2 | 0.97 | 0.28 | 0.45 |
| August | 3 | 1.19 | 0.26 | 0.36 |
| August | 4 | 3.11 | 0.19 | 0.04 |
| Total | n/a | 0.75 | 0.24 | 0.43 |