# Peer review of "Updated Arctic melt pond fraction dataset and trends 2002–2023 using ENVISAT and Sentinel-3 remote sensing data"

_The Cryosphere, 2023_

## Referee Comment (RC1)

**Summary**

This manuscript employs the MERIS melt pond retrieval algorithm on OLCI data. As MERIS and OLCI have almost same spectral response function, this transferability looks possible. However, there is still room for improvement in the research content of this paper. I recommend that the paper can be considered for publication after major revision.

**Specific comments**

I suggest dividing the results and discussion sections for better organization. Discussions are currently scattered throughout the paper.

I think there are spectral differences between MERIS and OLCI. The sensitive analysis should be done on the same targets (i.e., sea ice, melt pond, lead, and ocean) between MERIS and OLCI. Although MERIS and OLCI don't have same temporal period, similar targets can be used. This analysis will show a good example applying the algorithm for old satellite to a successor satellite.

P4, L118: Multiple satellite have been used for their purpose, but it is hard to follow. It would be good to add a table summarizing satellites used in this paper. Furthermore, a flowchart of this paper would enhance clarity.

P5, L146: In terms of cloud screening, cloud shadows are appeared on the sea ice surface depending on angles. Please describe the cloud shadow removal process if authors did.

Figure3 c: Please explain why sentinel-3 OLCI MPF produces 1 comparing to different sentinel-2 MSI MPF.

P8, L225: Please justify why the authors select these two cases. I think there are good cases in the 50 scenes. In the 50 scenes, some cases (i.e., leads and small open water) highly affect melt pond fraction showing diverse spectral behavior. Please add more diverse cases.

Figure5 c: Please explain why sentinel-3 OLCI MPF produces 1 comparing to different sentinel-2 MSI MPF.

P13, L330: I don't get it how this conclusion was reached.

P13, L357: Open water influence the retrieval of melt pond fraction. The leads and small open water surrounded by sea ice are also influence the retrieval of melt pond fraction. It would be good to mention this.

P14: While there is no map in the figure 9, the part 4 describes geographical information.

P17, 434-435: If sea ice type shift have progressed, it would be good to add melt onset data for this description.

Figure 11: Please demonstrate more about figure 11 in the paper.

4.2: The trend of Arctic sea ice concentration and thickness is steeper than long-term melt pond trend due to sea ice type shift above described?

Figure 12: It is difficult to see the many weekly trends. It would be good to show monthly trend instead of weekly with error bars.

**Technical corrections**

P9, L239: Figure 2a to Figure 3a?

P9, L242: level means level ice?

Figure7: IC means SIC?

What stands for OWF?

P13, L352: Please add this reference Rostosky et al., (2023) below.

P14, L367: data 2012-2016 not available should be mentioned.

Figure9: Please add some information about the thickness of blue and red color.

---

## Referee Comment (RC2)

**Review of tc-2023-142**

**Updated Arctic melt pond fraction dataset and trends 2002–2023 using ENVISAT and Sentinel-3 remote sensing data**

**By Istomina and others**

**General Comments**

The authors update a previously developed and published algorithm called MPD for the estimation of melt pond fraction (MPF) from satellite optical data, here showing its application to OLCI sensor data from Sentinel-3. The rationale is to provide continuity with the earlier dataset from the MERIS sensor aboard ENVISAT, which is no longer available, but technologically similar, making a 17-year dataset spanning 2002-present with a gap from 2012-2016. The quality of the dataset is evaluated in the current work by comparison to higher resolution optical data from Sentinel-2, to which a different MPF retrieval algorithm is applied (classification algorithm by Niehaus et al. (2023)). The MPD algorithm had been previously compared to ship-based, surface, and airborne data and as such its limitations are fairly well documented. The authors also do an analysis of trends in MPF, analyzing hemispheric trends and regional trends for the period 2002-present. Of note, an updated and more robust cloud screening technique is presented and applied to both the historic (MERIS) and current (OLCI) derived datasets.

Overall the MPD provides a reliable estimation of MPF based on assessment of agreement with other datasets, here (r=0.86) and in previous studies (Istomina et al., 2015). The characteristic nature of the MPD is to overestimate small MPF < 0.1 and underestimate large MPF >0.7. For the small MPF problem, in this paper the authors use spectral mixing simulations to effectively outline the limitations in the algorithm that are imposed by the sub-resolution mixture of surface types and especially the influence of open water. Using Sentinel-3 provides a consistent dataset that offers much potential for use in ocean-ice-atmosphere studies and for analyzing processes that might inform model parameterizations. MPF is also of concern to biological studies due to the importance of melt ponds on light transmission and primary production. The paper should be of interest to TC readership provided the authors address the comments provided here and by others.

1. The treatment of Sentinel-2 derived MPF as ground truth raises its own potential problems since this is another satellite derived dataset (although at higher spatial resolution) and is subject to its own retrieval limitations. Despite referencing the Sentinel-2 algorithm, the authors do not provide enough information on the algorithm and its limitations in the current paper to enable an effective intercomparison of the two products, or to build confidence that it should be assessed as a "truth". More detail needs to be provided. As well the Sentinel-2 MPF should not be identified as ground-truth.

2. Similarly, the comparison between Sentinel-2 and 3 done in Section 3.2 is too cursory in that it is a global comparison of the all overlapping data from the two datasets. The authors should provide a more detailed analysis that considers the ice condition (ice type) and some indication of the temporal component, i.e. how OLCI and MSI MPF compare in different seasonal phases of melt pond coverage when the spectral properties of the snow, ice, and ponds are different (but perhaps typical to some degree). In this context, is good performance of the MPD MPF product realized for first half of the melting season (June–July), as speculated on Line 360? The authors could use one or more of the already cited papers on melt pond evolution, to provide some structure to seasonal component of MPF evolution, e.g. as opposed to calendar months (Eicken et al., 2004; Polashenski et al., 2012). On the other hand, the spectral mixing analysis in Section 3.2 is lengthy and hard to follow in parts, and

should be improved. It important to be clear as to what observations from these results were used to make changes to the current MPD, versus what are being used to highlight possible error sources or identify areas for future iterations of the algorithm.

3. The authors use terminology regarding seasonal stage that isn't consistent with the literature, especially melt onset, which most often likely means pond onset i.e. formation of melt ponds that occurs some time after melt onset, when there is enough meltwater that flooding is possible. This will need to be addressed throughout the paper.

**Minor Comments**

L17-19: Clarify the ranges ("small" and "middle" are ambiguous).

L18: The snow would not be saturated if it is before melt onset. Is this supposed to pond onset?

L28: Again the term melt onset is confusing here as it was not analyzed, but the onset of ponds was, so perhaps "pond onset" is more appropriate.

L34: 2016-2023.

L35: "…world (Rantanen et al., 2022), the…" (delete "and")

L38: It would be better to say that it is due to the ocean being darker than the sea ice, not the other way around.

L43: What is meant by surface melt in the context of a sea ice ECV? How would it differ from the albedo, which changes due to melt?

L46-54: Briefly provide some detail on what is affecting the different remote sensing methods (e.g., PM emission is affected by the presence of open water from melt ponds on sea ice, etc.).

L51: "A PM based sea ice drift product…"

L55: "GCMs"

L59: "forecasts"

L61: Update the references (e.g. MOSAiC melt pond studies).

L62: Do you mean "climate conforming"? It is unclear.

L64: Use "MPF" instead of "melt pond" for consistency.

L69: delete "of"

L79: add space after "1.4 GHz,"

L80-84: The link between penetration depth and MPF estimation is unclear. Is it not more-so the emission differences between ice and melt pond that enable MPF retrieval, and the presence of open water along with melt ponds that confuses the MPF retrieval because melt pond and OW have similar emission characteristics (regardless of penetration depth)?

Line 86: Add "synthetic aperture radar"

L88: Melt pond and open water could have equally high backscatter in windy conditions.

L93: Clarify what Terra and Aqua are (platforms each with MODIS sensors).

L93: data "are"

L94: Start a new sentence at "important …"

L99-102: Sentence "In addition, …" is not clear and could probably be broken up into two sentences.

L103: "…a MPF dataset"

L105: "…of an earlier…"

L120-123: The information listed should be written out and presented more clearly or summarized in a table.

L134: "…and the absorption coefficient…"

L137-139: Provide some more information on the ice sampled here, given the importance of the dataset (types, melt pond conditions e.g. depth, etc.).

L142: "The MPD has been…."

L159: change to "swaths" and change "overflight" to "overpass"

L175: change to "swaths"

L183: acronym SIC should be defined earlier

L183: "Examples of the daily …."

L184: "…are shown"

L186: Use "SIC"

L189: See general comments. It is not correct to call the Sentinel-2 data "ground truth".

L190: In Fig. 1 it is confusing to have 0% MPF and no-data as both white color.

L202: Use italics for "*in situ*"

L205: "mixtures"

L210: Clarify what is meant by "within ice surface types".

L219: "MPF"

L229: As mentioned before, this should be pond onset not melt onset.

L242: "… a high fraction of ridges…"

L259-260: "darker water saturated sea ice" could be better described. Is this blue ice and/or optically thin ice?

L274: "…a translucent scattering"

L292: Clarify what is meant by typical MYI. There is MYI north of the Canadian Archipelago, even within the Archipelago, due to it ending up there after drifting in the area during summer. Depending on

where the ice transitioned to MYI before drifting to its imaged location, it may be typical (which also depends on the authors' definition of typical).

L316: Insert comma after Fig. 5

L317: "…in (Fig. 7)" doesn't need brackets around Fig. 7 as done on Line 320. Note on Line 323 the text "Figure 7" is used. Be consistent with style used for identifying figures in the text here and elsewhere.

L323: "…melt pond type…"

L330: "… favorable condition"

L332: "…MPD MPF"

L334: "… SIC is shown as color-coding of the data points"

L335: "…bright sea ice surfaces"

L338-343: It is hard to follow if these solutions are implemented in the current algorithm or not. Use of text "can still be accommodated" and "can be accounted for" makes it unclear if these ideas are for future consideration or not.

L343-344: pluralize "version" and use MPF for "melt pond fraction"

L355: There is not water saturated sea ice before melt.

L366-367: Used "averaged" and "analyzed" i.e. past tense for methods implemented.

L385-390: change descriptions of past methods to past tense

L397: See general comment about melt onset (above).

L451: Add spaces each side of "–"

L453: "A significance hemispheric…"

L454: "The last three weeks…"

L456: Add spaces each side of "–"

L483: "…for the middle MPF range"

L533: Is there a statement regarding the Sentinel-2 MPF dataset?

---

## Author Comment (AC1)

**Authors' response to the Referee 1 comments on tc-2023-142: Updated Arctic melt pond fraction dataset and trends 2002 – 2023 using ENVISAT and Sentinel-3 remote sensing data**

**by L. Istomina et al.**

The authors thank the reviewer for their valuable comments. We would like to highlight two points we felt were important upon evaluation of the revised manuscript:

1. Many of the reviewer's comments seem to point toward insufficient clarity of the text. We made sure to carefully check the text once again to make sure that all important information, like references to tables, figures, etc, is explicitly included and cannot be missed. We believe that the revised text will therefore be easier to follow.
2. The comments regarding the influence of open water onto the MPD retrieval were addressed in the parallel manuscript by Niehaus et al 2024, which was submitted in The Cryosphere later than the present manuscript, and for this reason was not referred to in the original text, but passed the review earlier and is now published. We now include this important reference and hope that now the present manuscript appears in the correct context, namely, the delivery of the long-term dataset and discussion on the quality of the two-surface (as opposed to three-surface) melt pond fraction retrieval.

In the following text, we address the comments of the reviewer 1 point by point, whereas the reviewer comment is highlighted with bold face font, and the authors' response follows in normal font.

**I suggest dividing the results and discussion sections for better organization. Discussions are currently scattered throughout the paper.**

We have carefully reconsidered the current organization of the manuscript and concluded that, given the large thematic spread of the aspects discussed in Sections 3.1, 3.2, 4.1 and 4.2, it would not be beneficial to separate the discussions from the section where the topic is introduced. The current paper presentation aims at highlighting the limitations of the MPD retrieval, so that the dataset users have no false understanding regarding its quality. In this context, we feel that separating the discussion into one section would 1) hinder the comprehensive understanding of each of the presented aspects, 2) render the discussion section – being now a mix of separate aspects – largely unusable and hard to follow. That's why we have decided to leave the manuscript structure unchanged, and instead take care to check the language for clarity to ensure better understanding.

**I think there are spectral differences between MERIS and OLCI. The sensitive analysis should be done on the same targets (i.e., sea ice, melt pond, lead, and ocean) between MERIS and OLCI. Although MERIS and OLCI don't have same temporal period, similar targets can be used. This analysis will show a good example applying the algorithm for old satellite to a successor satellite.**

The spectral resolution of the MERIS and OLCI sensors is summarized in Table 1. While some channels indeed differ, the 9 channels that are used for the MPD retrieval are exactly the same, so we do not expect MPF discrepancies stemming from the spectral resolution issue. As for the intercomparison of MERIS and OLCI on selected targets of given surface types: as the data gap between MERIS data (summer 2011) and OLCI data (summer 2017) is 5 years long, there is no possibility to select exactly the same surface types at a 300m spatial resolution. Should the surface selection be imperfect, any resulting MPF discrepancy cannot be exclusively attributed to the algorithm performance but rather to the difference in the MERIS and OLCI surface type. Thus, the authors refrain from such a comparison.

**P4, L118: Multiple satellite have been used for their purpose, but it is hard to follow. It would be good to add a table summarizing satellites used in this paper. Furthermore, a flowchart of this paper would enhance clarity.**

The sensors used in this paper have been summarized in Tables 1 and 2. We will add the reference to the Table 1 also in L. 119 to enhance clarity. Table 2 is already referred to in the original version of the text.

**P5, L146: In terms of cloud screening, cloud shadows are appeared on the sea ice surface depending on angles. Please describe the cloud shadow removal process if authors did.**

The snow/ice flag approach, described in Istomina et al., 2010 and used as a reference cloud mask in the presented approach, is based on locating the spectral behavior of snow and ice surfaces as appearing in the Arctic and screening out all other surfaces. This means, that clouds and cloud shadows are screened out automatically as they do not present the spectral signature of snow and ice. The corresponding clarification has been added to the text.

**Figure3 c: Please explain why sentinel-3 OLCI MPF produces 1 comparing to different sentinel-2 MSI MPF.**

This explanation is already contained in the original text on P.9, L. 236-239, right after the Figure 3c was referred to. This MPD behavior is caused by the spectral ambiguity of the TOA reflectances shown in Fig. 7, so that two solution families representing the same TOA reflectances equally well are used interchangeably, depending on whether the transport scattering coefficient is limiting the sea ice properties or not. This case highlights the problem of the spectral ambiguity, where ONE set of measured TOA reflectances corresponds to MANY different subpixel fractions of different surface types. Corresponding clarification will be added into the text.

**P8, L225: Please justify why the authors select these two cases. I think there are good cases in the 50 scenes. In the 50 scenes, some cases (i.e., leads and small open water) highly affect melt pond fraction showing diverse spectral behavior. Please add more diverse cases.**

The selection of the two cases presented in Fig. 2 and 4 stems from the need to illustrate the MPD performance on the entire span of MPF range, from low to very high MPFs of 100%, possibly showing the spectral ambiguity issue without the open water influence, being the simplest case. As can be seen from the text corresponding to Figure 2-5 on PP 8-10, a multitude of factors can affect the algorithm performance. The authors are convinced that the potential data users need to be aware of these details. Regarding the effect of the open water on the MPF: this effect has been illustrated on the entire dataset of 50 scenes in Fig. 6a and even in more detail, with the suggestion on how to improve the performance in the presence of open water, in Niehaus et al., 2024 (see preface to this author response above). The corresponding reference and clarification will be added in the next version of the text.

**Figure5 c: Please explain why sentinel-3 OLCI MPF produces 1 comparing to different sentinel-2 MSI MPF.**

Here again, like in the case of earlier mentioned Fig. 3c, two solution families are present due to spectral ambiguity issue, whereas the split between them is caused by the reached boundary criterium on the ice scattering coefficient. The corresponding clarification will be added to the text.

**P13, L330: I don't get it how this conclusion was reached.**

Due to the ambiguity of the spectral TOA reflectance measured by all moderate resolution spectroradiometers like MERIS, MODIS, OLCI, etc (shown in Fig. 7), the three-surface MPF retrieval will not be able to distinguish whether all 3 surfaces are present, and if, which of the surfaces are present. This is due to the fact, that the TOA measured signal is spectrally ambiguous, meaning, a multitude of surface combinations and fractions give *same* TOA reflectance, making the inverse retrieval from this

TOA reflectances to derive the subpixel surface fractions inaccurate. This means that, given no additional external information is applied, the 3-surface MPF retrieval will be always able to find a suitable combination of 3 surfaces even when only 2 surfaces are present, as the spectral TOA reflectance it obtains from the satellite data does not constrain the surface mixture confidently. Which of this many combinations it then mostly finds, depends on the training and calibration of the algorithm, but since the limited training data presents limited surface conditions, there will always be conditions which the 3-surface MPF retrieval without additional data is not able to retrieve correctly.

Niehaus et al., 2024, presents the 3-surface retrieval with additional data and addresses this issue in detail. This reference and a corresponding clarification have been added to the text.

**P13, L357: Open water influence the retrieval of melt pond fraction. The leads and small open water surrounded by sea ice are also influence the retrieval of melt pond fraction. It would be good to mention this.**

Indeed, here leads and other open water areas within sea ice are meant. Corresponding clarification will be added into the text.

**P14: While there is no map in the figure 9, the part 4 describes geographical information.**

Indeed, Figure 9 addresses hemispheric averages and investigates the uniformity of the dataset between the two sensors MERIS and OLCI. The geographical distribution of the melt pond fraction trend is shown in Fig. 10.

**P17, 434-435: If sea ice type shift have progressed, it would be good to add melt onset data for this description.**

The retrieval of the sea ice type in summer is not a trivial task, as passive microwave ice type retrievals are hindered by open water and melt pond presence. In this context, it is out of scope of the current manuscript to perform accurate ice type or melt onset retrievals to use as an evidence for the observed melt pond fraction trend behavior, where the melt onset data would be of course of importance. Nevertheless, we felt it was important to mention ice type shift as a potential reason already in this manuscript, to establish context for the future studies. Corresponding clarification will be added into the manuscript.

**Figure 11: Please demonstrate more about figure 11 in the paper.**

Figure 11 is explained on P. 15 L 389 onwards. Indeed, it can be referred more often in the Section 4.1, to point out areas where the MPF trend is significant. This will be done in the next version of the paper.

**4.2: The trend of Arctic sea ice concentration and thickness is steeper than long-term melt pond trend due to sea ice type shift above described?**

As remote sensing passive microwave sea ice concentration products are unreliable in summer due to the presence of open melt ponds and wet sea ice surface, there is currently no way to prove this fact at a global Actic scale. However, we felt it was important to mention that the thinning of the Arctic sea ice as shown by Sumata et al., 2023; Haas et al., 2008; can potentially cause negative sea ice concentration trend which might affect the MPF trends presented here.

**Figure 12: It is difficult to see the many weekly trends. It would be good to show monthly trend instead of weekly with error bars.**

The advantage of the daily melt pond fraction product as presented in this paper, in contrast to e.g. MODIS 8-day reflectance product which is also sometimes used for the MPF detection, is the high temporal resolution which is beneficial for both climate model input as well as independent melt pond

studies. While the authors agree that the presented Figure 12 cannot be easily compared to the above mentioned 8-day composites or monthly averages presented in other studies, it is out of scope of this manuscript to perform such comparisons, which will be done in the future. Other than easy comparisons to other MPF products, the authors could not see any advantage of giving up the high temporal resolution in this figure and decided to keep the weekly trends as presented in the original manuscript.

**Technical corrections**

**P9, L239: Figure 2a to Figure 3a?**

The reviewer probably suspects a typo here, but no, indeed, the Figure 2a lower square is meant as correctly written in the original version of the text.

**P9, L242: level means level ice?**

P9 L242 reads: "Here, the sea ice surface is not as level: high fraction of ridges… can be observed". In this sense, yes, it means the sea ice surface is not level and contains relief in form of ridges. The word "level" will be exchanged by "smooth" in the next version of the manuscript for more clarity.

**Figure7: IC means SIC? What stands for OWF?**

OWF stands for open water fraction and was defined on P11, before Fig. 7. IC stands for ice concentration and will be specified in the next version of the text.

**P13, L352: Please add this reference Rostosky et al., (2023) below.**

Here the reference to Rostosky and Spreen, 2023 is meant. This will be corrected in the next version of the paper.

**P14, L367: data 2012-2016 not available should be mentioned.**

Indeed, it can be mentioned also here, and will be added in the next version of the manuscript.

**Figure9: Please add some information about the thickness of blue and red color**

The Fig. 9 shows standard violin plots, where the histogram of the data is shown with the color thickness. The corresponding clarification will be added to the next version of the manuscript.

---

## Author Comment (AC2)

**Authors' response to the Referee 2 comments on tc-2023-142: Updated Arctic melt pond fraction dataset and trends 2002 – 2023 using ENVISAT and Sentinel-3 remote sensing data**

**by L. Istomina et al.**

The authors thank the referee for their valuable comments, which allow us to improve the manuscript and make the text so much more concise and readable. Please find our point-by-point response to the comments below. The reviewer comments are highlighted with bold face font, and the authors' response follows in a normal font.

**1. The treatment of Sentinel-2 derived MPF as ground truth raises its own potential problems since this is another satellite derived dataset (although at higher spatial resolution) and is subject to its own retrieval limitations. Despite referencing the Sentinel-2 algorithm, the authors do not provide enough information on the algorithm and its limitations in the current paper to enable an effective intercomparison of the two products, or to build confidence that it should be assessed as a "truth". More detail needs to be provided. As well the Sentinel-2 MPF should not be identified as ground truth.**

The authors agree that the Sentinel-2 comparison dataset is not an *in situ* measurement and therefore should not be referred to as "ground truth". We will correct this in the future version of the manuscript and use term "comparison dataset".

However, Sentinel-2 MPF stems from data where melt ponds can be clearly recognized by shape and in some cases even counted manually. In comparison to OLCI data where each pixel inevitably contains a subpixel surface mixture – a situation prone to complications as shown in Section 3.2, - Sentinel-2 MPF with the spatial resolution of 10 m offers a drastic increase of dataset quality. The spatial resolution of 10 meters together with unprecedented spatial coverage gives a valuable addition to the earlier presented validation effort (Istomina et al., 2015a) and thus allows us to significantly improve our understanding on the MPD algorithm performance as compared to the previously published validation (Istomina et al., 2015a). Therefore, the authors insist on presenting this comparison. Niehaus et al., 2023 presents successful validation and claims robustness of the retrieval used to produce the Sentinel-2 validation dataset. Although we agree that more details should be added on the Sentinel-2 dataset to aid better understanding of the manuscript and will do so in the future version of the manuscript, we do not see any faults in this peer-reviewed publication or in the Sentinel-2 MPF dataset which would prevent a quality comparison to OLCI MPF.

Details on the Sentinel-2 dataset will be added, as well as the "ground truth" expression will be exchanged throughout the manuscript.

**2. Similarly, the comparison between Sentinel-2 and 3 done in Section 3.2 is too cursory in that it is a global comparison of the all overlapping data from the two datasets. The authors should provide a more detailed analysis that considers the ice condition (ice type) and some indication of the temporal component, i.e. how OLCI and MSI MPF compare in different seasonal phases of melt pond coverage when the spectral properties of the snow, ice, and ponds are different (but perhaps typical to some degree). In this context, is good performance of the MPD MPF product realized for first half of the melting season (June–July), as speculated on Line 360? The authors could use one or more of the already cited papers on melt pond evolution, to provide some structure to seasonal component of MPF evolution, e.g. as opposed to calendar months (Eicken et al., 2004; Polashenski et al., 2012). On the other hand, the spectral mixing analysis in Section 3.2 is lengthy and hard to follow in parts, and should be improved. It important to be clear as to what observations from these results**

**were used to make changes to the current MPD, versus what are being used to highlight possible error sources or identify areas for future iterations of the algorithm.**

Indeed, the comparison of the entire Sentinel-2 dataset has been limited to gridded data only, to illustrate the performance issues of the MPD retrieval at a global scale. The more detailed comparison of the same dataset to MPD MPF on the original OLCI resolution is presented in Niehaus et al., 2024, which uses the Sentinel-2 dataset by Niehaus et al., 2023, to improve the MPD retrieval presented here and to include 3 surface retrieval (sea ice-melt pond-open water). The scope of this manuscript is not the improvement of the MPD algorithm, but MPD performance assessment in addition to existing validation and the long-term trends MERIS+OLCI. In the next version of the manuscript, we will take extra care to highlight this context and make sure that the references important to its understanding are included (Istomina et al., 2015; Niehaus et al., 2024).

Unfortunately, we cannot retrieve ice type or the phase of the melt evolution in this version of the MPD retrieval. Moreover, Section 3.2 particularly deals with the question why optical data from OLCI or MERIS (or MODIS) is not sufficient to perform such a retrieval, so that using melt phase instead of weekly trends is prone to high uncertainties and is not used. Instead, we adopt the already published (Istomina et al., 2015) approach to present weekly trends of the combined dataset, as a direct continuation of the earlier presented MERIS trends and as improvement on the past MERIS dataset. The newly presented MPF trends have changed significantly with the addition of the OLCI part of the dataset, and the authors like to preserve this comparison which might be of interest to the scientific community.

However, we agree that details on ice type and melt evolution can be beneficial for MPF retrieval. This approach is realized in the recently published companion paper Niehaus et al., 2024, which presents an improvement on the MPD retrieval. They include temperature history of a given ice parcel, so that more information on the ice type can be assumed. Their approach, however, cannot be directly applied to MERIS data, so no long-term trends are possible with the new 3-surface MPD retrieval by Niehaus et al., 2024. To obtain long-term trends from 2002 till present, the initially published version of the MPD retrieval (Istomina et al., 2015a; Zege et al., 2015) needs to be used, which is the focus of the presented manuscript.

In the next version of the manuscript, we will add the details on the temporal performance of the MPD retrieval and once again proof-read the Section 3.2 for readability. The above-mentioned details as well as reference to Niehaus et al., 2024, which has been submitted later than the current work and was therefore not yet included, will be added as well. Also, we will take extra caution to highlight that in the present manuscript, the MPD algorithm was not modified and is the same as presented in Istomina et al 2015, with the exception of a uniform cloud screening applied to both parts of the dataset. Therefore, the data comparison and the spectral mixing clarification in Section 3.2 as well as the examples shown in the Section 3.1 can be seen as performance assessment of the original 2-surface MPD retrieval, e.g. when the long-term MPF dataset is used as input to climate models.

**3. The authors use terminology regarding seasonal stage that isn't consistent with the literature, especially melt onset, which most often likely means pond onset i.e. formation of melt ponds that occurs some time after melt onset, when there is enough meltwater that flooding is possible. This will need to be addressed throughout the paper.**

The terminology regarding melt stages which used in the manuscript is largely adopted from Eicken et al., 2002. The authors agree that it should be updated and are grateful for this remark. Indeed, there are some cases where melt onset and onset of ponding have been used

interchangeably, which is of course inconsistent. We will proof-read the manuscript for consistency of usage in the next version of the manuscript.

**Minor Comments**

**L17-19: Clarify the ranges ("small" and "middle" are ambiguous).**

Here "small" means "0-0.2" and "middle" means "0.2-0.8". This will be changed in the next version of the manuscript.

**L18: The snow would not be saturated if it is before melt onset. Is this supposed to pond onset?**

Yes, thank you for highlighting this case. This will be changed to "pond onset" in the next version of the manuscript.

**L28: Again the term melt onset is confusing here as it was not analyzed, but the onset of ponds was, so perhaps "pond onset" is more appropriate.**

Yes, here the onset of ponding is meant. It will be changed accordingly in the next version of the manuscript.

**L34: 2016-2023.**

At the time of writing, the year 2023 was not yet finished. In the final version, this sentence will be updated accordingly.

**L35: "…world (Rantanen et al., 2022), the…" (delete "and")**

Will be deleted in the next version of the manuscript.

**L38: It would be better to say that it is due to the ocean being darker than the sea ice, not the other way around.**

Will be changed accordingly in the next version of the manuscript.

**L43: What is meant by surface melt in the context of a sea ice ECV? How would it differ from the albedo, which changes due to melt?**

The melt pond fraction is not directly linked to the sea ice albedo, as same fraction of melt pond can have very different albedo depending on the pond type, or better, on the thickness of the underlying sea ice. So that in an extreme case there may be a 100% melt pond fraction on thick ice, with albedo so high that it is similar to 0% melt pond fraction on a wet, large grain sea ice, which is in turn darker than the regular sea ice. Similar albedo, but very different melt situation. Therefore, MPF ECV data should ideally be separated from the sea ice albedo ECV.

**L46-54: Briefly provide some detail on what is affecting the different remote sensing methods (e.g., PM emission is affected by the presence of open water from melt ponds on sea ice, etc.).**

Indeed, we will provide corresponding physical context on how summer conditions affect PM SIC, altimeter-based SIT and PM sea ice drift retrievals, in the next version of the manuscript.

**L51: "A PM based sea ice drift product…"**

Will be changed accordingly in the next version of the manuscript.

**L55: "GCMs"**

Will be changed accordingly in the next version of the manuscript.

**L59: "forecasts"**

Will be changed accordingly in the next version of the manuscript.

**L61: Update the references (e.g. MOSAiC melt pond studies).**

The following references on MOSAIC melt pond studies will be added:

Webster et al., 2022; Light et al., 2022.

**L62: Do you mean "climate conforming"? It is unclear.**

Here the datasets of global coverage and high quality which can be used as climatology datasets are meant. Will exchanged for "high quality" in the next version of the manuscript.

**L64: Use "MPF" instead of "melt pond" for consistency.**

"MPF" will be used at this point in the next version of the manuscript.

**L69: delete "of"**

Will be deleted in the next version of the manuscript.

**L79: add space after "1.4 GHz,"**

Will be added into the next version of the manuscript.

**L80-84: The link between penetration depth and MPF estimation is unclear. Is it not more-so the emission differences between ice and melt pond that enable MPF retrieval, and the presence of open water along with melt ponds that confuses the MPF retrieval because melt pond and OW have similar emission characteristics (regardless of penetration depth)?**

The penetration depth is essential here because the underlying sea ice in the pond ("pond bottom") cannot be recognized due to low penetration depth, that is, melt pond cannot be distinguished from open water. This will be added into the next version of the manuscript.

**Line 86: Add "synthetic aperture radar"**

Will be added into the next version of the manuscript.

**L88: Melt pond and open water could have equally high backscatter in windy conditions.**

Indeed, this is also the case and will be added into the text. Still, the point stands: melt ponds and open water cannot be reliably distinguished using SAR only backscattered signal (but could be potentially distinguished using shape and image analysis).

**L93: Clarify what Terra and Aqua are (platforms each with MODIS sensors).**

The clarification will be added to the manuscript.

**L93: data "are"**

Indeed, this should read "MODIS data are…". We will correct this.

**L94: Start a new sentence at "important …"**

The sentences will be separated in the next version of the text.

**L99-102: Sentence "In addition, ..." is not clear and could probably be broken up into two sentences.**

The sentences will be split in the next version of the manuscript.

**L103: "...a MPF dataset"**

Of course, this typo will be corrected in the future version.

**L105: "...of an earlier..."**

Currently the article "an" is missing, it will be added in the future version of the text.

**L120-123: The information listed should be written out and presented more clearly or summarized in a table.**

Reference to Table 1 will be added also in L. 119 and the text on L.120-123 will be checked for clarity.

**L134: "...and the absorption coefficient..."**

Of course, we will add "and" into the text at this point.

**L137-139: Provide some more information on the ice sampled here, given the importance of the dataset (types, melt pond conditions e.g. depth, etc.).**

The PANGAEA dataset referred to in this sentence contains spectra of bare ice of various grain sizes, snow, dark and light melt ponds with or without the ice lid, blue ice without the scattering layer, for a range of ice thickness from 30cm to 2.5m within the melt ponds. These details will be added into the future version of the manuscript.

**L142: "The MPD has been...."**

We will exchange "have" for "has" in this sentence.

**L159: change to "swaths" and change "overflight" to "overpass"**

Will be changed in the next version of the manuscript.

**L175: change to "swaths"**

Will be changed in the next version of the manuscript.

**L183: acronym SIC should be defined earlier**

Indeed, we will define the sea ice concentration – SIC – on P2 L47.

**L183: "Examples of the daily ...."**

Of course, as there are two examples, we will use multiple here.

**L184: "...are shown"**

Also this will be corrected, corresponding to multiple examples as mentioned above.

**L186: Use "SIC"**

Will be used in the next version of the manuscript.

**L189: See general comments. It is not correct to call the Sentinel-2 data "ground truth".**

This will be corrected here and throughout the manuscript.

**L190: In Fig. 1 it is confusing to have 0% MPF and no-data as both white color.**

We do not have any MPD data with exactly (numerically) 0% MPF, therefore this choice of the colorbar.

**L202: Use italics for "in situ"**

Will be done in the next version of the manuscript.

**L205: "mixtures"**

Will be used in the next version of the manuscript.

**L210: Clarify what is meant by "within ice surface types".**

Here the main feature of the melting ice as opposed to white ice, namely the decreased near infrared reflectance, is exploited, thus making it possible to only use a ratio of two channels to, in the first approximation, separate melting and non-melting sea ice. This clarification will be added into the manuscript.

**L219: "MPF"**

Will be used in the next version of the manuscript.

**L229: As mentioned before, this should be pond onset not melt onset.**

Will be changed in the next version of the manuscript.

**L242: "... a high fraction of ridges..."**

Will be used in the next version of the manuscript.

**L259-260: "darker water saturated sea ice" could be better described. Is this blue ice and/or optically thin ice?**

It can be both thin ice or blue ice, or subnivean ponds, we do not have information to distinguish these surfaces as the spectral ambiguity plot in Fig. 7 is pointing out. This clarification will be added at this point of the manuscript.

**L274: "...a translucent scattering"**

Will be corrected in the next version of the manuscript.

**L292: Clarify what is meant by typical MYI. There is MYI north of the Canadian Archipelago, even within the Archipelago, due to it ending up there after drifting in the area during summer. Depending on where the ice transitioned to MYI before drifting to its imaged location, it may be typical (which also depends on the authors' definition of typical).**

Typical MYI in the sense that it is rougher than the FYI, so that this increase in sea ice relief will lead to decrease of the maximal possible MPF on this sea ice. We will clarify this in the next version of the text.

**L316: Insert comma after Fig. 5**

Will be done in the next version of the manuscript.

**L317: "…in (Fig. 7)" doesn't need brackets around Fig. 7 as done on Line 320. Note on Line 323 the text "Figure 7" is used. Be consistent with style used for identifying figures in the text here and elsewhere.**

We will check the style for consistency.

**L323: "…melt pond type…"**

Indeed, here only one melt pond type is meant – dark melt pond – this will be explained in the next version of the paper.

**L330: "… favorable condition"**

Line 330 reads as intended "…, thus biasing an otherwise favorable for the MPF retrieval situation". No condition is meant here. We will correct this sentence for more clarity: "…, thus biasing even this favorable MPF situation of two surface classes only."

**L332: "…MPD MPF"**

The double MPF will be corrected to MPD MPF as advised.

**L334: "… SIC is shown as color-coding of the data points"**

Thank you for this suggestion, we will reformulate the original sentence to the advised version.

**L335: "…bright sea ice surfaces"**

We will pluralize "surface" in the next version of the text.

**L338-343: It is hard to follow if these solutions are implemented in the current algorithm or not. Use of text "can still be accommodated" and "can be accounted for" makes it unclear if these ideas are for future consideration or not.**

Indeed, these are ideas for future consideration which have been already realized in the companion publication by Niehaus et al. 2024. At the time of writing, the paper by Niehaus et al. 2024 has not yet been submitted. But now, as it has been already published, we will include the reference to Niehaus et al., 2024 and correct the text here for more clarity.

**L343-344: pluralize "version" and use MPF for "melt pond fraction"**

We will do so in the next version of the paper.

**L355: There is not water saturated sea ice before melt.**

Here the onset of ponding is meant, will be corrected in the future version of the manuscript.

**L366-367: Used "averaged" and "analyzed" i.e. past tense for methods implemented.**

We will implement past tense here.

**L385-390: change descriptions of past methods to past tense**

Also here, past tense will be implemented.

**L397: See general comment about melt onset (above).**

The reviewer means the stages of melt occurring on FYI, with the first stage of melt being "1. Melt onset" and being ambiguous. We suggest to correct it like following:

"1. Melt onset followed be onset of ponding". This brings the most clarity, although deviates from the original Eicken et al., 2002 formulation.

**L451: Add spaces each side of "–"**

Will be added into the next version of the manuscript.

**L453: "A significance hemispheric…"**

Indeed, we will insert the article "A" in the text at this point.

**L454: "The last three weeks…"**

Here as well, we will insert the missing article "the" in the text.

**L456: Add spaces each side of "–"**

Will be added into the next version of the manuscript.

**L483: "…for the middle MPF range"**

The missing article "the" will be inserted at this point.

**L533: Is there a statement regarding the Sentinel-2 MPF dataset?**

Indeed, this has been overlooked here and will be included in the next version of the text.

---

## Author Response (AR2)

Point by point response to the second review of the TC-2023-142

**Reviewer #1 has requested a few more very minor items need to be addressed. They are:**
**1. Please add a flowchart of this paper to enhance a clarity.**

*The manuscript flowchart has been added as Figure 1. The corresponding sentence at line 117 has been added as well, and the figure numbers have been changed correspondingly throughout the manuscript.*

**2. Please add this Niehaus et al., 2024 in the reference lists.**

The Nieahaus et al., 2024 reference has been added at lines 729-731.

**Reviewer #1 also stated: I found "will be added in the next version of the text" throughout the response. These will be added next revision round?**

*Initially, the point-by-point response to the reviewers was prepared before the manuscript has been changed, that's why we refer to these changes in future form. The version of the manuscript submitted after the point-by-point response had therefore already contained the declared changes. The authors agree that this formulation was unfortunate, and use the past tense formulation in the current point-by-point response instead.*